# META-MODEL-BASED META-POLICY OPTIMIZATION

## ABSTRACT

Model-based reinforcement learning (MBRL) has been applied to meta-learning settings and has demonstrated its high sample efficiency. However, in previous MBRL for meta-learning settings, policies are optimized via rollouts that fully rely on a predictive model of an environment. Thus, its performance in a real environment tends to degrade when the predictive model is inaccurate. In this paper, we prove that performance degradation can be suppressed by using branched meta-rollouts. On the basis of this theoretical analysis, we propose Meta-Model-based Meta-Policy Optimization (M3PO), in which the branched meta-rollouts are used for policy optimization. We demonstrate that M3PO outperforms existing meta reinforcement learning methods in continuous-control benchmarks.

## 1 INTRODUCTION

Reinforcement learning (RL) methods have achieved remarkable success in many decision-making tasks, such as playing video games or controlling robots (e.g., Gu et al. (2017); Mnih et al. (2015)). In conventional RL methods, when multiple tasks are to be solved, a policy is independently learned for individual tasks. In general, each learning requires millions of training samples from the environment. This independent learning with a large number of samples prevents conventional RL methods from being applied to practical multi-task problems (e.g., robotic manipulation problems involving grasping or moving different types of objects (Yu et al., 2019)). Meta-learning methods (Schmidhuber et al., 1996; Thrun & Pratt, 1998) have recently gained much attention as a promising solution to this problem (Finn et al., 2017). They learn a structure shared in the tasks by using a large number of samples collected across the parts of the tasks. Once learned, these methods can adapt quickly to new (or the rest of the) tasks with a small number of samples given.

Meta-RL methods have previously been introduced into both model-free and model-based settings. For model-free settings, there are two main types of approaches proposed so far, recurrent-based policy adaptation (Duan et al., 2017; Mishra et al., 2018; Rakelly et al., 2019; Wang et al., 2016) and gradient-based policy adaptation (Al-Shedivat et al., 2018; Finn & Levine, 2018; Finn et al., 2017; Gupta et al., 2018; Rothfuss et al., 2019; Stadie et al., 2018). In these approaches, policies adapt to a new task by leveraging the history of past trajectories. Following previous work (Clavera et al., 2018), we refer to these adaptive policies as **meta-policies** in our paper. In these model-free meta-RL methods, in addition to learning control policies, the learning of policy adaptation is also required (Mendonca et al., 2019). Thus, these methods require more training samples than conventional RL methods.

For model-based settings, there have been relatively few approaches proposed so far. Sæmundsson et al. (2018) and Perez et al. (2020) use a predictive model (i.e., a transition model) conditioned by a latent variable for model predictive control. Nagabandi et al. (2019a;b) introduced both recurrent-based and gradient-based meta-learning methods into model-based RL. In these approaches, the predictive models adapt to a new task by leveraging the history of past trajectories. In analogy to the meta-policy, we refer to these adaptive predictive models as **meta-models** in our paper. Generally, these model-based meta-RL approaches are more sample efficient than the model-free approaches. *However, in these approaches, the meta-policy (or the course of actions) is optimized via rollouts relying fully on the meta-model. Thus, its performance in a real environment tends to degrade when the meta-model is inaccurate.* In this paper, we address this performance degradation problem in model-based meta-RL.

After reviewing related work (Section 2) and preliminaries (Section 3), we present our work by first formulating model-based meta-RL (Section 4). Model-based (and model-free) meta-RL settings have typically been formulated as special cases of solving partially observable Markov decision processes (POMDPs) (e.g.,Duan et al. (2017); Killian et al. (2017); Perez et al. (2020)). In these special cases, specific assumptions, such as intra-episode task invariance, are additionally introduced. However, there are model-based meta-RL settings where such assumptions do not hold (e.g., Nagabandi et al. (2019a;b)). To include these settings into our scope, we formulate model-based meta-RL settings as solving POMDPs without introducing such additional assumptions. Then, we conduct theoretical analysis on its performance guarantee (Section 5). We first analyse the performance guarantee in full meta-model-based rollouts, which most of the previous model-based meta-RL methods hold. We then introduce the notion of branched meta-rollouts. Branched meta-rollouts are Dyna-style rollouts (Sutton, 1991) in which we can adjust the reliance on the meta-model and real environment data. We show that the performance degradation due to the meta-model error in the branched meta-rollouts is smaller than that in the full meta-model-based rollouts. On the basis of this theoretical analysis, we propose a practical model-based meta-RL method called Meta-Model-based Meta-Policy Optimization (M3PO) where the meta-model is used in the branched rollout manner (Section 6). Finally, we experimentally demonstrate that M3PO outperforms existing methods in continuous-control benchmarks (Section 7).

We make the following contributions in both theoretical and empirical frontiers. **Theoretical frontier: 1**. Our work is the first attempt to provide a theoretical relation between learning the meta-model and the real environment performance. In the aforementioned model-based meta-RL literature, it has not been clear how learning the meta-model relates to real environment performance. Our theoretical analysis provides relations between them (Theorems 1, 2 and 3). This result theoretically justifies meta-training a good transition model to improve overall performance in the real environment. **2**. Our analysis also reveals that the use of branched meta-rollouts can suppress performance degradation due to meta-model errors. **3**. We refine previous fundamental theories proposed by Janner et al. (2019) to consider important premises more properly (Theorems 4 and 5). This modification is important to strictly guarantee the performance especially when the model-rollout length is long. **Empirical frontier:** We propose and show the effectiveness of M3PO. Notably, we show that M3PO achieves better sample efficiency than existing meta-RL methods in complex tasks, such as controlling humanoids.

## 2 RELATED WORK

In this section, we review related work on POMDPs and theoretical analysis in model-based RL.

**Partially observable Markov decision processes** [1]**:** In our paper, we formulate model-based meta-RL as solving POMDPs, and provide its performance guarantee under the branched meta-rollout scheme. POMDPs are a long-studied problem (e.g., (Ghavamzadeh et al., 2015; Sun, 2019; Sun et al., 2019)), and many works have discussed a performance guarantee of RL methods to solve POMDPs. However, the performance guarantee of the RL methods based on branched meta-rollouts has not been discussed in the literature. On the other hand, a number of researchers (Igl et al., 2018; Lee et al., 2019; Zintgraf et al., 2020) have proposed model-free RL methods to solve a POMDP without prior knowledge of the accurate model. However, they do not provide theoretical analyses of performance. In this work, by contrast, we propose a model-based meta-RL method and provide theoretical analyses on its performance guarantee.

**Theoretical analysis on the performance of model-based RL:** Several theoretical analyses on the performance of model-based RL have been provided in previous work (Feinberg et al., 2018; Henaff, 2019; Janner et al., 2019; Luo et al., 2018; Rajeswaran et al., 2020). In these theoretical analyses, standard Markov decision processes (MDPs) are assumed, and the meta-learning (or POMDP) setting is not discussed. In contrast, our work provides a theoretical analysis on the meta-learning (and POMDP) setting, by substantially extending the work of Janner et al. (2019). Specifically, Janner et al. (2019) analysed the performance guarantee of branched rollouts on MDPs, and introduced branched rollouts into a model-based RL algorithm. We extend their analysis and algorithm to a meta-learning (POMDP) case. In addition, we modify their theorems so that important premises

---

[1]We include works on Bayes-adaptive MDPs (Ghavamzadeh et al., 2015; Zintgraf et al., 2020) because they are a special case of POMDPs.

(e.g., the effect of multiple-model rollout factors) are more properly considered. See A.1 in the appendix for a more detailed discussion of our contribution.

## 3 PRELIMINARIES

**Meta reinforcement learning:** We assume online adaptation situations (Nagabandi et al., 2019a;b) where the agent can leverage a few samples to adapt to a new task. Here, a task specifies the transition probability and the reward function. Information about task identity cannot be observed by the agent, and the task may change at any step in an episode. A meta-RL process is composed of meta-training and meta-testing. In meta-training, a policy and a predictive model that are prepared for efficient adaptation are learned with a meta-training task set. In meta-testing, on the basis of the meta-training result, the policy and the predictive model adapt to a new task. For the adaptation, the trajectory observed from the beginning of the episode to the current time step is leveraged. As we noted earlier, we call an adaptive policy and a predictive model of this sort a meta-policy and a meta-model, respectively.

**Partially observable Markov decision processes:** We formalize our problem with a POMDP, which is defined as a tuple $\langle \mathcal{O}, \mathcal{S}, \mathcal{A}, p_{\mathrm{ob}}, r, \gamma, p_{\mathrm{st}} \rangle$. Here, $\mathcal{O}$ is a set of observations, $\mathcal{S}$ is a set of hidden states, $\mathcal{A}$ is a set of actions, $p_{\mathrm{ob}} := \mathcal{O} \times \mathcal{S} \times \mathcal{A} \to [0,1]$ is the observation probability, $p_{\mathrm{st}} := \mathcal{S} \times \mathcal{S} \times \mathcal{A} \to [0,1]$ is the state transition probability, $r : \mathcal{S} \times \mathcal{A} \to \mathbb{R}$ is a reward function and $\gamma \in [0,1)$ is a discount factor. At time step $t$, these functions are used as $p_{\mathrm{st}}(s_t|s_{t-1}, a_{t-1})$, $p_{\mathrm{ob}}(o_t|s_t, a_{t-1})$ [2] and $r_t = r(s_t, a_t)$. The agent cannot directly observe the hidden state, but receives the observation instead. The agent selects an action on the basis of a policy $\pi := p(a_{t+1}|h_t)$. Here, $h_t$ is a history (the past trajectories) defined as $h_t := \{a_0, o_0, ..., a_t, o_t\}$. We denote the set of the histories by $\mathcal{H}$. Given the definition of the history, the history transition probability can be defined as $p(h_{t+1}|a_{t+1}, h_t) := p(o_{t+1}|h_t)$. Here, $p(o_{t+1}|h_t) := \sum_{s_{t+1}} \sum_{s_t} p(s_t|h_t)p(s_{t+1}|s_t, a_t)p(o_{t+1}|s_{t+1}, a_t)$, where $p(s_t|h_t)$ is the belief about the hidden state. The goal of RL in the POMDP is to find the optimal policy $\pi^*$ that maximizes the expected return $R := \sum_{t=0}^{\infty} \gamma^t r_t$ (i.e., $\pi^* = \arg \max_{\pi} \mathbb{E}_{a \sim \pi, h \sim p}[R]$).

## 4 FORMULATING MODEL-BASED META-RL

In this section, we formulate model-based meta-RL as solving a POMDP by using a parameterized meta-policy and a meta-model. The outline of our formulation is shown in Figure 4 in the appendix.

In our formulation, the task is included in the hidden state: $\mathcal{S} := \mathcal{T} \times \mathcal{S}'$. Here $\mathcal{T}$ is the set of task $\tau$ and $\mathcal{S}'$ is the set of the other hidden state factors $s'$. With this definition, the state transition probability, observation probability and reward function can be defined respectively as follows: $p(s_{t+1}|s_t, a_t) = p(\tau_{t+1}, s'_{t+1}|\tau_t, s'_t, a_t)$, $p(o_{t+1}|s_{t+1}, a_t) = p(o_{t+1}|\tau_{t+1}, s'_{t+1}, a_t)$ and $r(s_t, a_t) = r(\tau_t, s'_t, a_t)$. In addition, as with Finn & Levine (2018); Finn et al. (2017); Rakelly et al. (2019), we assume that the task set $\mathcal{T}$ and the initial task distribution $p(\tau_0)$ do not change in meta-training and meta-testing. Owing to this assumption, in our analysis and algorithm, meta-training and meta-testing can be seen as an identical one. Note that the task can be changed during an episode. Namely, the value of $\tau_{t+1}$ is not necessarily equal to that of $\tau_t$.

We define a meta-policy and meta-model as $\pi_\phi(a_{t+1}|h_t)$ and $p_\theta(r_t, o_{t+1}|h_t)$, respectively. Here, $\phi$ and $\theta$ are learnable parameters for them. $r_t$ and $o_{t+1}$ are assumed to be conditionally independent given $h_t$, i.e., $p_\theta(r_t, o_{t+1}|h_t) = p_\theta(r_t|h_t) \cdot p_\theta(o_{t+1}|h_t)$. As with $p(h_{t+1}|a_{t+1}, h_t)$, the meta-model for the history can be defined as $p_\theta(h_{t+1}|a_{t+1}, h_t) := p_\theta(o_{t+1}|h_t)$.

We use the parameterized meta-model and meta-policy as shown in Algorithm 1. This algorithm is composed of 1) data collection in the real environment and 2) optimization of the meta-policy and meta-model. In 1), the data is collected from the real environment with the meta-policy and stored into a dataset $\mathcal{D}$. In 2), the meta-policy and meta-model are optimized to maximize $\mathbb{E}_{a \sim \pi_\phi, r, h \sim p_\theta}[R] - C(\epsilon_m(\theta), \epsilon_\pi(\phi))$. Here, $\mathbb{E}_{a \sim \pi_\phi, r, h \sim p_\theta}[R]$ is a meta-model return (the return of

---

[2] For simplicity, we use these probabilities by abbreviating the subscripts "st" and "ob."

---

**Algorithm 1** Abstract Meta Model-based Meta-Policy Optimization (Abstract M3PO)

---

1: Initialize meta-policy $\pi_\phi$, meta-model $p_\theta$ and dataset $\mathcal{D}$.
2: **for** $N$ epochs **do**
3:     Collect trajectories from environment in accordance with $\pi_\phi$: $\mathcal{D} = \mathcal{D} \cup \{(h_t, o_{t+1}, r_t)\}$.
4:     Optimize $\pi_\phi$ and $p_\theta$: $(\phi, \theta) \leftarrow \underset{(\phi,\theta)}{\arg\max} \mathbb{E}_{a\sim\pi_\phi, r, h\sim p_\theta}[R] - C(\epsilon_m(\theta), \epsilon_\pi(\phi))$. Here, $\mathcal{D}$ is
    used to evaluate $\epsilon_m(\theta)$ and $\epsilon_\pi(\phi)$ and generate an initial history $h_0$.
5: **end for**

---

the meta-policy on the meta-model) [3]. $C(\epsilon_m(\theta), \epsilon_\pi(\phi))$ is a discrepancy depending on the two error quantities $\epsilon_m$ and $\epsilon_\pi$. Their detailed definitions are introduced in the next section.

Our POMDP-based formulation covers a wide range of meta-RL settings including Bayes-adaptive MDPs (Zintgraf et al., 2020), hidden parameter-MDPs (Killian et al., 2017; Perez et al., 2020), parameterized MDPs (Duan et al., 2017; Finn & Levine, 2018), and the settings considered by Nagabandi et al. (2019a;b) and Jabri et al. (2019). These settings can be primarily recovered in our formulation by introducing both or either of the following assumptions (Asm1 and Asm2) [4]. **Asm1:** the task does not change during an episode. **Asm2:** observation $o$ is identical to the hidden state $s'$: $o = s'$. The Bayes-adaptive MDPs, hidden-parameter MDPs and parameterized MDPs can be recovered by introducing both Asm1 and Asm2. The setting of Nagabandi et al. (2019a;b) can be recovered by introducing Asm2, and the setting of Jabri et al. (2019) can be recovered by introducing Asm1. As a detailed example, we recover the parameterized MDPs in Appendix A.5.

## 5   PERFORMANCE GUARANTEES OF MODEL-BASED META-RL

In this section, we analyse the performance guarantee of model-based meta-RL with an inaccurate meta-model. In Section 5.1, we provide the performance guarantee in a full meta-model-based rollout. In Section 5.2, we introduce the notion of a branched meta-rollout and analyse its performance guarantee. We show that the meta-model error is less harmful in the branched meta-rollout than that in the full meta-model-rollout.

### 5.1   PERFORMANCE GUARANTEE IN A FULL META-MODEL-BASED ROLLOUT CASE

Our goal is to outline a theoretical framework in which we can provide performance guarantees for Algorithm 1. To show the guarantees, we construct a lower bound taking the following form:

$$\mathbb{E}_{\pi_\phi, p}[R] \geq \mathbb{E}_{\pi_\phi, p_\theta}[R] - C(\epsilon_m(\theta), \epsilon_\pi(\phi)). \tag{1}$$

Here, $\mathbb{E}_{\pi_\phi, p}[R]$ denotes a true return (i.e., the return of the meta-policy in the real environment). The discrepancy between these returns, $C$, can be expressed as the function of two error quantities: the generalization error of the meta-policy and the distribution shift due to the updated meta-policy. For our analysis, we define the bounds of the generalization error $\epsilon_m$ and the distribution shift $\epsilon_\pi$ as follows:

**Definition 1.** $\epsilon_m(\theta) := \max_t \mathbb{E}_{a\sim\pi_\mathcal{D}, h\sim p}[D_{TV}(p(h_{t+1}|a_{t+1}, h_t)||p_\theta(h_{t+1}|a_{t+1}, h_t))]$. *Here,* $D_{TV}$ *is a total variation distance and* $\pi_\mathcal{D}$ *is the data-collection policy whose actions contained in* $\mathcal{D}$ *follow* [5].

**Definition 2.** $\epsilon_\pi(\phi) := \max_{h_t} D_{TV}(\pi_\mathcal{D}(a_{t+1}|h_t)||\pi_\phi(a_{t+1}|h_t))$.

We also assume that the expected reward is bounded by a constant $r_{max}$.

**Definition 3.** $r_{max} > \max_t \left| \sum_{s_t} p(s_t|h_t) r(s_t, a_t) \right|$.

Now we present our bound, which is an extension of the theorem proposed in Janner et al. (2019).

---

[3] For simplicity, we use the abbreviated style $\mathbb{E}_{\pi_\phi, p_\theta}[R]$.

[4] For simplicity, we omit implementation-level assumptions (e.g., "meta-policy or meta-model are implemented on the basis of gradient-based MAML" in Finn & Levine (2018); Nagabandi et al. (2019a)).

[5] As with Janner et al. (2019), to simplify our analysis, we assume that the meta-model can accurately estimate reward. We discuss the case in which the reward prediction of a meta-model is inaccurate in A.8.

**Theorem 1** (The POMDP extension of Theorem 4.1. in Janner et al. (2019))**.** *Let* $\epsilon_m = \max_t \mathbb{E}_{a \sim \pi_{\mathcal{D}}, h \sim p} \left[ D_{TV} \left( p(h_{t+1}|a_{t+1}, h_t) || p_\theta(h_{t+1}|a_{t+1}, h_t) \right) \right]$ *and* $\epsilon_\pi = \max_{h_t} D_{TV} \left( \pi_{\mathcal{D}}(a_{t+1}|h_t) || \pi_\phi(a_{t+1}|h_t) \right)$. *Then, the true returns are bounded from below by meta-model returns of the meta-policy and discrepancy:*

$$\mathbb{E}_{\pi_\phi, p}[R] \geq \mathbb{E}_{\pi_\phi, p_\theta}[R] - \underbrace{r_{max} \left[ \frac{2\gamma(\epsilon_m + 2\epsilon_\pi)}{(1-\gamma)^2} + \frac{4\epsilon_\pi}{(1-\gamma)} \right]}_{C(\epsilon_m(\theta), \epsilon_\pi(\phi))} \quad (2)$$

This theorem implies that the discrepancy of the returns under full meta-model-based rollout scales linearly with both $\epsilon_m$ and $\epsilon_\pi$. If we can reduce the discrepancy $C$, the two returns are closer to each other. As a result, the performance degradation is more significantly suppressed. In the next section, we discuss a new meta-model usage to reduce the discrepancy induced by the meta-model error $\epsilon_m$.

### 5.2 PERFORMANCE GUARANTEE IN THE BRANCHED META-ROLLOUT CASE

The analysis of Theorem 1 relies on running full rollouts through the meta-model, causing meta-model errors to compound. This is reflected in the bound by a factor scaling quadratically with the effective horizon, $1/(1-\gamma)$. In such cases, we can improve the algorithm by choosing to rely less on the meta-model and instead more on real environment data.

To allow for adjustment between meta-model-based and model-free rollouts, we introduce the notion of **a branched meta-rollout**. The branched meta-rollout is a kind of Dyna-style rollout (Sutton, 1991), in which the meta-model-based rollout is run as being branched from real environment data. More concretely, the rollout is run in the following two processes. **1)** We begin a rollout from a history under the data-collection meta-policy's history distribution $p_{\pi_{\mathcal{D}}}(h_t)$, and **2)** we then run $k$ steps in accordance with $\pi_\phi$ under the learned meta-model $p_\theta$.

Under such a scheme, the true return can be bounded from below:

**Theorem 2.** *Under the $k$ steps branched meta-rollouts, using the bound of a meta-model error under $\pi_{\mathcal{D}}$, $\epsilon_m = \max_t \mathbb{E}_{a \sim \pi_{\mathcal{D}}, h \sim p, t} \left[ D_{TV} \left( p(h'|h, a) || p_\theta(h'|h, a) \right) \right]$, the bound of the meta-policy shift $\epsilon_\pi = \max_{h_t} D_{TV} \left( \pi_{\mathcal{D}} || \pi_\phi \right)$, and the return on the meta-model $\mathbb{E}_{(a,h) \sim \mathcal{D}_{model}}[R]$ where $\mathcal{D}_{model}$ is the set of samples collected through branched rollouts, the following inequation holds,*

$$\mathbb{E}_{\pi_\phi, p}[R] \geq \mathbb{E}_{(a,h) \sim \mathcal{D}_{model}}[R] - r_{max} \left\{ \frac{1+\gamma^2}{(1-\gamma)^2} 2\epsilon_\pi + \frac{\gamma - k\gamma^k + (k-1)\gamma^{k+1}}{(1-\gamma)^2} (\epsilon_\pi + \epsilon_m) \right.$$
$$\left. + \frac{\gamma^k - \gamma}{\gamma - 1} (\epsilon_\pi + \epsilon_m) + \frac{\gamma^k}{1-\gamma} (k+1)(\epsilon_\pi + \epsilon_m) \right\}. \quad (3)$$

The discrepancy factors relying on $\epsilon_m$ in Theorem 2 can be smaller than those relying on $\epsilon_m$ in Theorem 1 [6]. This indicates that the performance degradation due to the meta-model error can be more suppressed than that in the full meta-model-based rollout [7].

## 6 META-MODEL-BASED META-POLICY OPTIMIZATION WITH DEEP RL

In the previous section, we show that the use of branched meta-rollouts can suppress performance degradation. In this section, on the basis of this result, we modify Algorithm 1 so that the meta-policy and meta-model are optimized with $\mathbb{E}_{(a,h) \sim \mathcal{D}_{model}}[R] - C(\epsilon_m(\theta), \epsilon_\pi(\phi))$, instead of with $\mathbb{E}_{\pi_\phi, p_\theta}[R] - C(\epsilon_m(\theta), \epsilon_\pi(\phi))$.

More specifically, we propose the following modifications to Algorithm 1:

**Meta-policy optimization:** The meta-policy is optimized with the branched meta-rollouts return $\mathbb{E}_{(a,h) \sim \mathcal{D}_{model}}[R]$ [8]. For the optimization, we adapt PEARL (Rakelly et al., 2019) be-

---

[6]See Corollary 1 in the appendix.

[7]In A.6, we also prove that the discrepancy can be further reduced by introducing an additional assumption.

[8]To stabilize the learning, we omit $C(\epsilon_m(\theta), \epsilon_\pi(\phi))$ from the optimization objective for the meta-policy. The transition of $\epsilon_\pi(\theta)$ during learning with this limited optimization objective is shown in Figure 12 in the appendix. The result indicates that $\epsilon_\pi(\theta)$ tends to decrease as the training data size (epoch) grows.

---

**Algorithm 2** Meta-Model-based Meta-Policy Optimization with Deep RL (M3PO)

---

1: Initialize meta-policy $\pi_\phi$, meta-model $p_\theta$, environment dataset $\mathcal{D}_{\text{env}}$, meta-model dataset $\mathcal{D}_{\text{model}}$.

2: **for** $N$ epochs **do**
3:   Train meta-model $p_\theta$ with $\mathcal{D}_{\text{env}}$: $\theta \leftarrow \arg\max_\theta \mathbb{E}_{\mathcal{D}_{\text{env}}} \left[ p_\theta(r_t, o_{t+1}|h_t) \right]$
4:   **for** $E$ steps **do**
5:     Take actions according to $\pi_\phi$; add the trajectory to $\mathcal{D}_{\text{env}}$
6:     **for** $M$ model rollouts **do**
7:       Sample $h_t$ uniformly from $\mathcal{D}_{\text{env}}$
8:       Perform $k$-step meta-model rollouts starting from $h_t$ using meta-policy $\pi_\phi$; add fictitious trajectories to $\mathcal{D}_{\text{model}}$
9:     **end for**
10:     **for** $G$ gradient updates **do**
11:       Update policy parameters with $\mathcal{D}_{\text{model}}$: $\phi \leftarrow \phi - \nabla_\phi J_{\mathcal{D}_{\text{model}}}(\phi)$
12:     **end for**
13:   **end for**
14: **end for**

---

cause it achieved a good learning performance in meta-learning settings. We use the fictitious trajectories generated from the branched meta-rollouts to optimize the meta-policy. Formally, $\pi_\phi$ is optimized by using the gradient of optimization objective $J_{\mathcal{D}_{\text{model}}}(\phi) := \mathbb{E}_{(a,h)\sim\mathcal{D}_{\text{model}}} \left[ D_{KL} \left( \pi_\phi || \exp \left( Q_{\pi_\phi} - V_{\pi_\phi} \right) \right) \right]$. Here, $D_{KL}$ is the Kullback-Leibler divergence, $Q_{\pi_\phi}(a_{t+1}, h_t) := \mathbb{E}_{(r,h)\sim\mathcal{D}_{\text{model}}} \left[ R|a_{t+1}, h_t \right]$ and $V_{\pi_\phi}(h_t) := \sum_{a_{t+1}} Q_{\pi_\phi}(a_{t+1}, h_t) \pi_\phi(a_{t+1}|h_t)$. As in PEARL, in $\pi_\phi$, the latent context is estimated by using past trajectories. The estimated context is then used to augment the policy input. Formally, the meta-policy is implemented as $\pi_\phi(a_{t+1}|h_t) = \sum_z \pi_\phi(a_{t+1}|o_t, z) p_\phi(z|a_0, o_0, ..., a_t, o_t)$. $z$ is the latent context and $p_\phi(z|a_0, o_0, ..., a_t, o_t)$ is a context encoder. $\pi_\phi(a_{t+1}|o_t, z)$ is the conditioned policy. Similarly, in $Q_{\pi_\phi}$ and $V_{\pi_\phi}$, the estimated latent context is also used to augment their input.

**Meta-model optimization:** The meta-model is optimized to minimize the discrepancy (i.e., minimize $\epsilon_m$) [9]. For the meta-model, to consider both aleatoric and epistemic uncertainties, we use a bootstrap ensemble of $B$ dynamics models $\{p_\theta^1, ..., p_\theta^B\}$. Here, $p_\theta^i$ is the $i$-th conditional Gaussian distribution with diagonal covariance: $p_\theta^i(r_t, o_{t+1}|h_t) = \mathcal{N} \left( r_{t+1}, o_{t+1}|\mu_\theta^i(h_t), \sigma_\theta^i(h_t) \right)$. $\mu_\theta^i$ and $\sigma_\theta^i$ are the mean and standard deviation, respectively. In our implementation, we use the recurrent-based architecture inspired by Duan et al. (2017); Nagabandi et al. (2019b); Rakelly et al. (2019); at each evaluation of the model, $\{a_1, o_1, ..., a_{t-1}, o_{t-1}\}$ in $h_t$ is fed to a recurrent neural network (RNN), and then its hidden unit output and $\{a_t, o_t\}$ in $h_t$ are fed to the feed-forward neural network that outputs the mean and standard deviation of the Gaussian distribution. We use the gated recurrent unit (GRU) (Cho et al., 2014) for the RNN. The recurrent layer in the GRU is composed of five sigmoid units. In addition, the feed-forward neural network is composed of an input layer, two hidden layers, and an output layer. The input and hidden layers are composed of 400 swish units (Ramachandran et al., 2017). To minimize $\epsilon_m$, we learn the meta-model via maximum likelihood estimation with $\mathcal{D}_{\text{env}}$. As in Lakshminarayanan et al. (2017), the ensemble $\{p_\theta^1, ..., p_\theta^B\}$ is learned on the shuffle of the real trajectories in $\mathcal{D}_{\text{env}}$. We apply a re-parameterization trick to evaluate the distributions, and the meta-model parameters $\theta$ are optimized with a gradient-based optimizer. For the gradient-based optimizer we use Adam (Kingma & Ba, 2015). To avoid overfitting, we use weight decay and early termination (Bishop, 2006).

The resulting algorithm is shown in Algorithm 2. The modifications "Meta-model optimization" and "Meta-policy optimization" in the above paragraph are reflected in line 3 and lines 4–13, respectively. In line 4, $k$-step branched meta-rollouts are run. The appropriately set $k$ contributes to decreasing the discrepancy, and suppresses performance degradation. Thus, we treat $k$ as a hyperparameter, and set it to different values in different environments so that the discrepancy decreases. For the experiments described in the next section, we tune the hyperparameters for this algorithm by a grid search. The search result for the hyperparameter values is described in Table 1 in the appendix.

---

[9]The transition of $\epsilon_m$ over training epochs is shown in Figure 11 in the appendix, and it indicates that the model error tends to decrease as the number of epochs increases.

For our experiments, we implemented our algorithm by extending the codebase of Model-based Policy Optimization [10]. We made two main extensions: (1) introduce a latent context to the policy as in PEARL and (2) replace a predictive model with the meta-model based on the RNN.

## 7  EXPERIMENTS

In this section, we report our experiments [11] aiming to answer the following questions: **Q.1:** Can our method (M3PO) outperform existing meta-RL methods? **Q.2:** How do meta-model-rollout lengths $k$ affect the actual performance?

In our experiments, we compare our method (M3PO) with two baseline methods: **PEARL** (Rakelly et al., 2019) and **Learning to adapt (L2A)** (Nagabandi et al., 2019a). More detailed information is described in A.10 in the appendix. We conduct a comparative evaluation of the methods on a variety of simulated robot environments using the MuJoCo physics engine (Todorov et al., 2012). We prepare the environments proposed in the meta-RL (Finn & Levine, 2018; Nagabandi et al., 2019a; Rakelly et al., 2019; Rothfuss et al., 2019) and robust-RL (Hiraoka et al., 2019; Rajeswaran et al., 2017) literature: **Halfcheetah-fwd-bwd**, **Halfcheetah-pier**, **Ant-fwd-bwd**, **Ant-crippled-leg**, **Walker2D-randomparams** and **Humanoid-direc**. In the environments, the agent is required to adapt to a fluidly changing task that the agent cannot directly observe. Detailed information about each environment is described in A.11 in the appendix.

Regarding Q1, our experimental results indicate that M3PO outperforms existing meta-RL methods. In Figure 1, the learning curves of M3PO and existing meta-RL methods (L2A and PEARL) on meta-training phases are shown. These learning curves indicate that the sample efficiency of M3PO is better than those of L2A and PEARL [12]. The performance (return) of L2A remains poor and does not improve even when the training data increases. PEARL can improve meta-policy performance via training in all environments. However, the degree of improvement of PEARL is smaller than that of M3PO. In a number of the environments (e.g., Halfcheetah-pier), the relative performance of M3PO against PEARL becomes asymptotically worse. This indicates that, as with Nagabandi et al. (2018), dynamic switching from M3PO to PEARL or other model-free approaches needs to be considered to further improve overall performance.

Regarding Q2, we conducted an evaluation of M3PO by varying its model-rollout length $k$. The evaluation results (Figure 2) indicate that the performance tends to degrade when the model-rollout length is long. We can see significant performance degradation especially in Ant-fwd-bwd and Humanoid-direc. In Ant-fwd-bwd, the performance at $k = 100$ is significantly worse than that at $k = 10$. In Humanoid-direc, the performance at $k = 5$ is significantly worse than that at $k = 1$. As we have seen, the performance degradation in Humanoid-direc is more sensitive to the model-rollout length than that in Ant-fwd-bwd. One reason for this is that the meta-model error in Humanoid-direc is larger than that in Ant-fwd-bwd (Figure 11 in the appendix).

An example of meta-policies learned by M3PO with 200k samples in Humanoid-direc is shown in Figure 3, and it indicates the learned policy successfully adapts to different tasks. Additional examples of meta-policy learned by PEARL and M3PO are shown in the video at the following link: `https://drive.google.com/file/d/1DRA-pmIWnHGNv5G_gFrml8YzKCtMcGnu/view?usp=sharing`

## 8  CONCLUSION

In this paper, we analysed the performance guarantee (and performance degradation) of MBRL in a meta-learning setting. We first formulated model-based reinforcement learning in a meta-learning setting as solving a POMDP. We then conducted theoretical analyses on the performance guarantee in both the full model-based rollout and the branched meta-rollout. We showed that the performance degradation due to the meta-model error in the branched meta-rollout is smaller than that in the full

---

[10] `https://github.com/JannerM/mbpo`

[11] The source code to replicate the experiments will be open to the public.

[12] Note that, in an early stage of the training phase, there are many test episodes in which unseen tasks appear. Therefore, the improvement of M3PO over L2A and PEARL at the early stage of learning indicates its high adaptation capability for unseen situations.

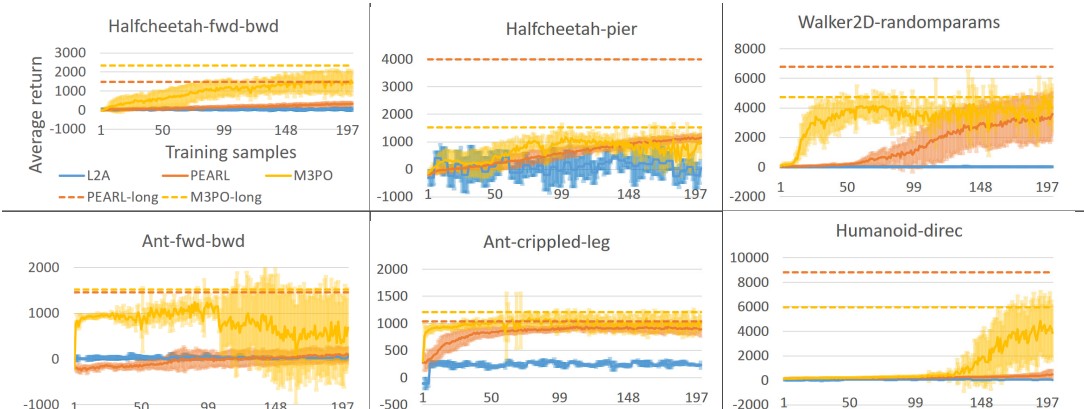

**Figure 1:** Learning curves. In each figure, the vertical axis represents returns, and the horizontal axis represents numbers of training samples (x1000). The meta-policy and meta-model are fixed and evaluated in terms of their average return on 50 episodes at every 5000 training samples for L2A and 1000 training samples for the other methods. In each episode, the task is initialized and changed randomly. Each method is evaluated in six trials, and average returns on the 50 episodes is further averaged over the trials. The averaged returns and their standard deviations are plotted in the figures. We also plot the best performances of PEARL and M3PO trained for a longer-term as PEARL-long and M3PO-long. Learning curves of PEARL and M3PO in the longer-term training are shown in Figure 13.

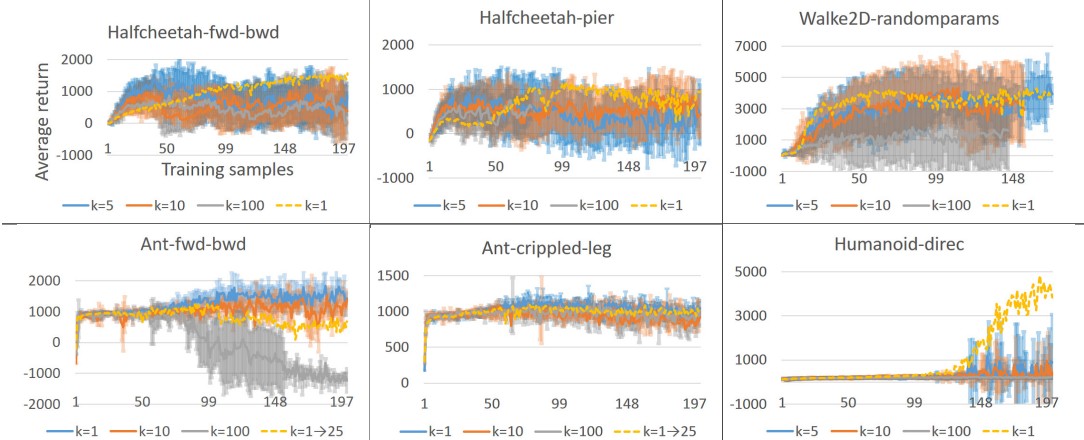

**Figure 2:** Learning curves of M3PO. In each figure, the vertical axis represents returns, and the horizontal axis represents numbers of training samples (x1000). The meta-policy and meta-model are fixed and evaluated their expected returns on 50 episodes per 1000 training samples. We run experiments by varying the rollout length $k$ of M3PO. In these experiments, the values of hyperparamters except $k$ are the same as those described in Table 1. Each case is evaluated in six trials, and the average return on the 50 episodes is further averaged over the trials. The averaged expected returns and their standard deviations are plotted in the figures. Dashed lines represent M3PO's learning curves in Figure 1. $k = x \rightarrow y$ means $k$ values linearly changes from $x$ to $y$ during leaning.

meta-model-based rollouts. Based on the theoretical result, we introduced branched meta-rollouts to policy optimization and proposed M3PO. Our experimental results show that it achieves better sample efficiency than PEARL and L2A.

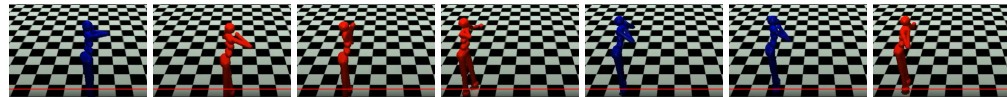

**Figure 3:** Example of a meta-policy learned by M3PO with 200k samples in Humanoid-direc.The humanoid is highlighted in accordance with a task (red: move to left, and blue: move to right). The figures are time-lapse snapshots, and the first figure on the left is the snapshot at the beginning time. Figures show the policy successfully adapting to the task change.

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

# A    APPENDICES

## A.1    HOW DOES OUR WORK DIFFER FROM JANNER ET AL. (2019)?

Although our work is grounded primarily on the basis of Janner et al. (2019), we provide non-trivial contributions in both theoretical and practical frontiers: **(1)** We provide theorems about the relation between true returns and returns on inaccurate predictive models (model returns) on a "meta-learning (POMDPs)" setting (Section 5). In their work, they provide theorems about the relation between the true returns and the model returns in the branched rollout in MDPs. In contrast, we provide theorems about the relation between the true returns and of the model returns in the branched rollout in POMDPs. *In addition, in the derivation of theorems (Theorems 4.2 and 4.3) in their work, a number of important premises are not properly taken into consideration* (the detailed discussion on it is described in the second paragraph in A.7). We provide new theorems, in which the premises are more properly reflected, for both MDPs and POMDPs (A.3, A.6, and A.7). **(2)** We extend the model-based policy optimization (MBPO) proposed by Janner et al. into the meta-learning (POMDPs) setting (Section 6). MBPO is for the MDP settings and does not support POMDP settings, while our method (M3PO) supports POMDP settings. Furthermore, we empirically demonstrate the usefulness of the meta-model usage in the branched rollout manner in the POMDP settings (Section 7).

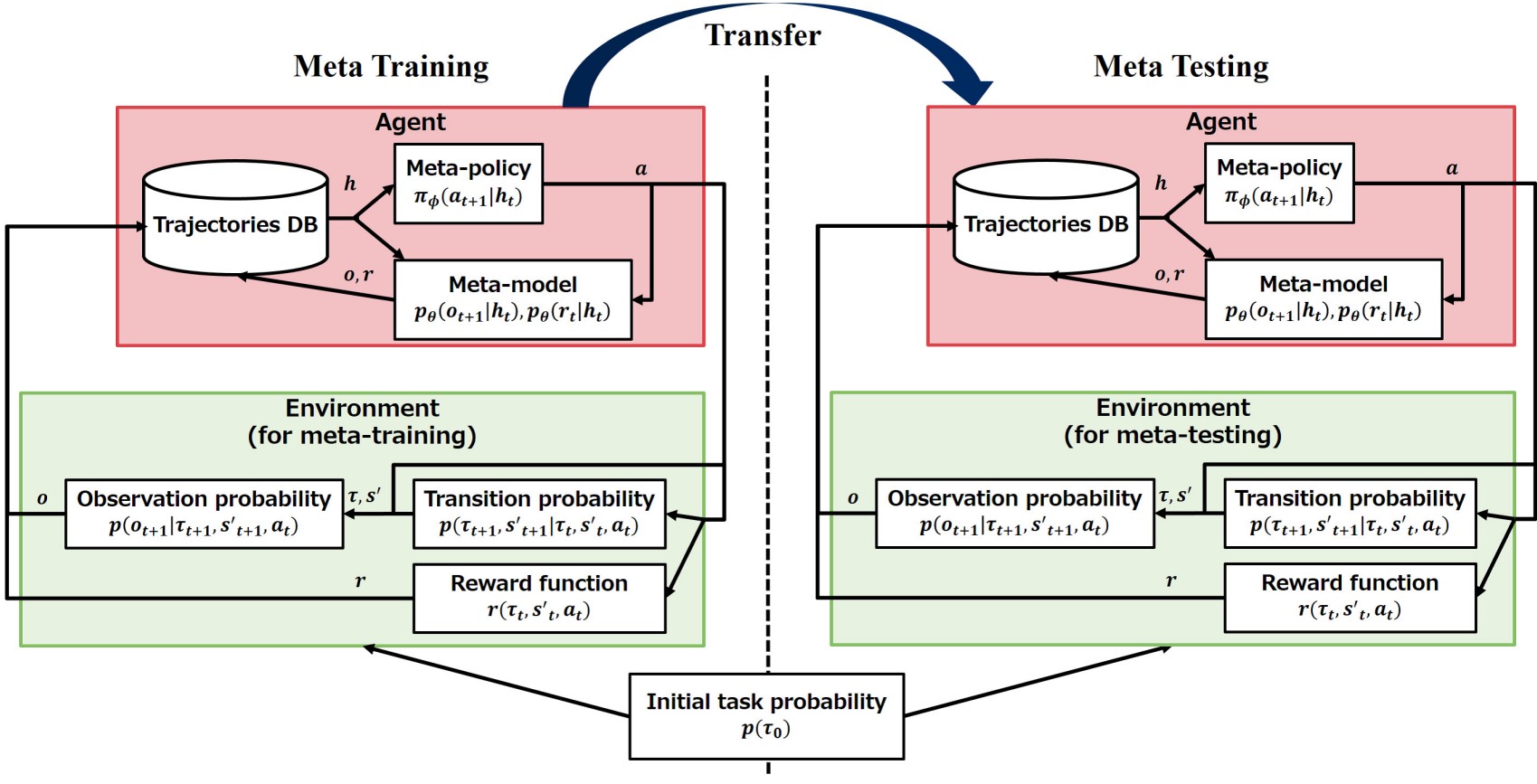

**Figure 4:** The outline of Model-based meta-RL formulation. Here, $\tau$ is a task, $s'$ is hidden state factors, $o$ is an observation, $h$ is a history, $a$ is an action and $r$ is a reward.

### A.3 PROOFS OF THEOREMS IN THE MAIN CONTENT

Before starting the derivation of the main theorems, we introduce a lemma useful for bridging POMDPs and MDPs.

**Lemma 1** (Silver & Veness (2010)). *Given a POMDP* $\langle \mathcal{O}, \mathcal{S}, \mathcal{A}, p_{ob}, r, \gamma, p_{st} \rangle$, *consider the derived MDP with histories as states,* $\langle \mathcal{H}, \mathcal{A}, \gamma, \bar{r}, p_{hi} \rangle$, *where* $\forall t.$ $p_{hi} := p(h_{t+1}|a_{t+1}, h_t) = \sum_{s_t \in \mathcal{S}} \sum_{s_{t+1} \in \mathcal{S}} p(s_t|h_t)p(s_{t+1}|s_t, a_t)p(o_{t+1}|s_{t+1}, a_t)$ *and* $\bar{r}(h_t, a_t) := \sum_{s_t \in S} p(s_t|h_t)r(s_t, a_t)$. *Then, the value function* $\bar{V}_\pi(h_t)$ *of the derived MDP is equal to the value function* $V_\pi(h_t)$ *of the POMDP.*

*Proof.* The statement can be derived by backward induction on the value functions. See the proof of Lemma 1 in Silver & Veness (2010) for details. □

PROOF OF THEOREM 1:

*Proof.* By Lemma 1, our problem in POMDPs can be mapped into the problem in MDPs, and then Theorem 4.1 in Janner et al. (2019) can be applied to the problem. □

Similarly, the proof of Theorem 2 is derived by mapping our problem into that in MDPs by Lemma 1 and leveraging theoretical results on MDPs.

PROOF OF THEOREM 2:

*Proof.* By Lemma 1, our problem in POMDPs can be mapped into the problem in MDPs, and then Theorem 4 in A.7 can be applied to the problem. □

### A.4 THE DISCREPANCY RELYING ON THE META-MODEL ERROR IN THEOREM 1 AND THAT IN THEOREM 2

**Corollary 1.** *The discrepancy factors relying on* $\epsilon_m$ *in Theorem 1,* $C_{Th1,m}$, *are equal to or larger than those relying on* $\epsilon_m$ *at* $k = 1$ *in Theorem 2,* $C_{Th2,m}$.

*Proof.* By Theorem 1 and 2,

$$C_{\text{Th1},m} = r_{\max} \frac{2\gamma \epsilon_m}{(1 - \gamma)^2}. \tag{4}$$

$$C_{\text{Th2},m} = r_{\max} \frac{\gamma}{1 - \gamma} 2\epsilon_m. \tag{5}$$

Given that $\gamma \in [0, 1)$, $r_{\max} > 0$ and $\epsilon_m \geq 0$,

$$C_{\text{Th1},m} - C_{\text{Th2},m} = r_{\max} \frac{4\gamma \epsilon_m - 2\gamma^2 \epsilon_m}{(1 - \gamma)^2} \tag{6}$$
$$\geq 0.$$

□

### A.5 CONNECTION TO A TYPICAL META-RL SETTING

In Section 4, we formulate model-based meta-RL to solve a POMDP. However, this formulation may make it difficult for certain readers to comprehend the connection to a typical meta-RL setting. Although in the normative meta-RL setting (the parameterized MDPs (Finn et al., 2017)), the objective to be optimized is given as the return expected with respect to a task distribution, such an objective does not appear in the formulation in Section 4. In this section, we show that such an objective can be derived by specializing the formulation in Section 4 under a number of assumptions

(Corollary 2). Then, we explain why we did not adopt such a specialization and maintained a more abstract formulation in Section 4.

First, letting a task set and a task distribution be denoted by $\mathcal{T}$ and $p(\tau)$ where $\tau \in \mathcal{T}$, respectively, we introduce the following assumptions:

**Assumption 1.** $\mathcal{S} := \mathcal{O} \times \mathcal{T}$.

**Assumption 2.** $p(s_{t+1}|s_t, a_t) := p(o_{t+1}|o_t, \tau_t, a_t) \cdot \mathbf{1}(\tau_{t+1} = \tau_t)$.

**Assumption 3.** For $t > 0$, $p(s_t|h_t) := p(\tau_t|h_t) \cdot \mathbf{1}(\tau_t = \tau_0)$.

**Assumption 4.** $p(\tau_0|h_0) := p(\tau_0)$.

Here, $\mathbf{1}(\cdot)$ is the indicator function that returns one if the argument is true, and zero otherwise.

With these assumptions, the following corollary holds:

**Corollary 2.** *Given a POMDP $\langle \mathcal{O}, \mathcal{S}, \mathcal{A}, p_{ob}, r, \gamma, p_{st} \rangle$ and a task set $\mathcal{T}$, consider the parameterized MDP with histories as states, $\langle \mathcal{H}, \mathcal{A}, \gamma, \bar{\bar{r}}, \bar{p}_{ob} \rangle$, where $\forall t.\ \bar{p}_{ob} := p(o_{t+1}|o_t, \tau_0, a_t)$ and $\bar{\bar{r}} := r(o_t, \tau_0, a_t)$. Under Assumptions 2~5, the expected return on the parameterized MDP $\mathbb{E}_{a\sim\pi, h\sim p, \tau\sim p(\tau)}[\sum_t^\infty \gamma^t \bar{\bar{r}}_t] := \sum_{\tau \in \mathcal{T}} p(\tau) \mathbb{E}_{a\sim\pi, h\sim p}[\sum_t^\infty \gamma^t \bar{\bar{r}}_t | \tau]$ is equal to the expected return on the POMDP $\mathbb{E}_{a\sim\pi, h\sim p}[R]$.*

*Proof.* By applying Lemma 1, the value function in a POMDP $\langle \mathcal{O}, \mathcal{S}, \mathcal{A}, p_{ob}, r, \gamma, p_{st} \rangle$ can be mapped to the value function $\bar{V}_\pi(h_t)$ in the derived MDP, which is $\langle \mathcal{H}, \mathcal{A}, \gamma, \bar{r}, p_{hi} \rangle$, where $\forall t.\ p_{hi} := p(h_{t+1}|a_{t+1}, h_t) = \sum_{s_t \in \mathcal{S}} \sum_{s_{t+1} \in \mathcal{S}} p(s_t|h_t) p(s_{t+1}|s_t, a_t) p(o_{t+1}|s_{t+1}, a_t)$ and $\bar{r}(h_t, a_t) := \sum_{s_t \in S} p(s_t|h_t) r(s_t, a_t)$.

By applying the assumptions, this value function can be transformed to a different representation that explicitly contains $\tau$ and its distribution:
For $t > 0$,

$$
\begin{aligned}
\bar{V}_\pi(h_t) =& \sum_{a_{t+1}} \pi(a_{t+1}|h_t) \Bigg\{ \sum_{s_t \in S} p(s_t|h_t) r(s_t, a_t) \\
&+ \gamma \sum_{o_{t+1} \in \mathcal{O}} \sum_{s_t \in \mathcal{S}} \sum_{s_{t+1} \in \mathcal{S}} p(s_t|h_t) p(s_{t+1}|s_t, a_t) p(o_{t+1}|s_{t+1}, a_t) \bar{V}_\pi(h_{t+1}) \Bigg\} \\
=& \sum_{a_{t+1}} \pi(a_{t+1}|h_t) \Bigg\{ \underbrace{r(o_t, \tau_0, a_t)}_{\bar{\bar{r}}_t} + \gamma \sum_{o_{t+1} \in \mathcal{O}} \underbrace{p(o_{t+1}|o_t, \tau_0, a_t)}_{\bar{p}_{ob}} \bar{V}_\pi(h_{t+1}) \Bigg\}.
\end{aligned} \tag{7}
$$

Likewise, for $t = 0$,

$$
\begin{aligned}
\bar{V}_\pi(h_0) =& \sum_{a_1} \pi(a_1|h_0) \Bigg\{ \sum_{\tau_0} p(\tau_0) r(o_0, \tau_0, a_0) + \gamma \sum_{o_1 \in \mathcal{O}} \sum_{\tau_0} p(\tau_0) p(o_1|o_0, \tau_0, a_0) \bar{V}_\pi(h_1) \Bigg\} \\
=& \sum_{\tau_0} p(\tau_0) \underbrace{\sum_{a_1} \pi(a_1|h_0) \Bigg\{ r(o_0, \tau_0, a_0) + \gamma \sum_{o_1 \in \mathcal{O}} p(o_1|o_0, \tau_0, a_0) \bar{V}_\pi(h_1) \Bigg\}}_{\bar{\bar{V}}_\pi(h_0)}.
\end{aligned} \tag{8}
$$

Therefore,

$$
\begin{aligned}
\mathbb{E}_{a \sim\pi, h\sim p}[R] =& \sum_{h_0} p(h_0) \bar{V}_\pi(h_0) \\
=& \sum_{\tau_0} p(\tau_0) \sum_{h_0} p(h_0) \bar{\bar{V}}_\pi(h_0) \\
=& \mathbb{E}_{a\sim\pi, h\sim p, \tau\sim p(\tau)}\left[ \sum_t^\infty \gamma^t \bar{\bar{r}}_t \right]
\end{aligned} \tag{9}
$$

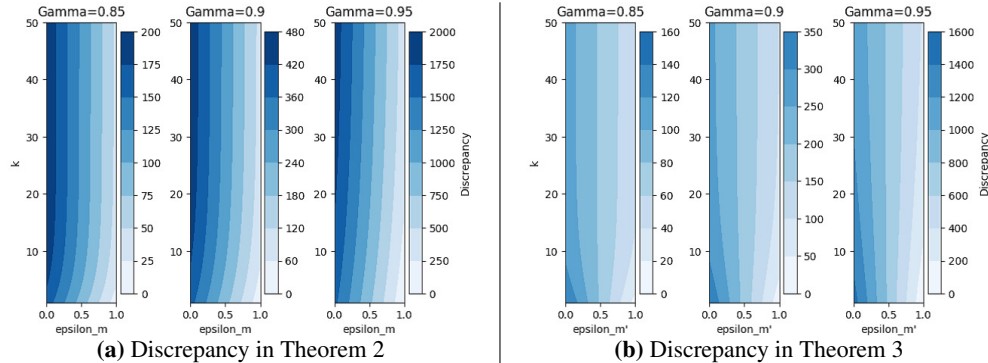

**(a)** Discrepancy in Theorem 2        **(b)** Discrepancy in Theorem 3

**Figure 5:** Discrepancy between $\mathbb{E}_{\pi_\phi, p}[R]$ and $\mathbb{E}_{(a,h) \sim \mathcal{D}_{\text{model}}}[R]$ in Theorem 2 and that in Theorem 3. Each vertical axis represents $k \in [1, 50]$ and each horizontal axis represents $\epsilon_m, \epsilon_{m'} \in [0, 1]$. In all figures, for evaluating the discrepancy values, we set the other variables as $r_{\max} = 1$, $\epsilon_\pi = 1 - \epsilon_m$ for (a), and $\epsilon_\pi = 1 - \epsilon_{m'}$ for (b). A key insight with the figures is following two points. First, discrepancy values in Theorem 2 (a) tend to increase as the k value increases, whereas those in Theorem 3 (b) are not. Thus, there could be an optimal $k$ value that larger than one. Second, the contour colours in (b) are paler than those in (a). This implies, given $\epsilon_m = \epsilon_{m'}$, Theorem 3 provides a tighter bound of the performance.

$\square$

By Corollary 2, our formulation in Section 4 can be specialized into a problem where the objective to be optimized is given as the return expected with respect to a task distribution. We can derive the meta-model returns with discrepancies for bounding the true return (i.e., $\mathbb{E}_{a \sim \pi_\phi, h \sim p_\theta, \tau \sim p(\tau)} \left[ \sum_t^\infty \gamma^t \bar{\bar{r}}_t \right] - C(\epsilon_m, \epsilon_\pi)$) by using Corollary 2 instead of Lemma 1 in the proofs of Theorem 1 and replacing $p_\theta(o_t | o_0, \tau_0, a_t)$ with $p_\theta(o_t | h_{t-1})$.

The main reason that we do not adopt such a specialization in Section 4 is to avoid restrictions induced by the assumptions (Assumption 2~5). For example, Assumption 2 states that hidden states are composed by observations and a task. Since observations can be observed from an agent by its definition, the information in the hidden states that cannot be observed from an agent is the task. However, in many meta-RL settings (e.g., application domains (Jabri et al., 2019) where video images are used as the observation), there should be other information that cannot be observed from the agent. In addition, Assumptions 3 and 4 state that the task is invariant in each episode. However, in many meta-RL settings (e.g., Nagabandi et al. (2019a;b)), the task can change to a different one in each episode. To avoid such restrictions, we decided not to specialize the formulation by introducing the assumptions.

### A.6 Additional analysis of the performance degradation in the branched meta-rollout

Figure 5a shows that the discrepancy value in Theorem 2 tends to monotonically increase as the value of $k$ increases, regardless of the values of $\gamma$ and $\epsilon_m$. This means that the optimal value of $k$ is always 1. However, intuitively, we may expect that there is an optimal value for $k$ that is higher than 1 when the meta-model error is small. As will be shown in this section, this intuition is correct. One of the main reasons for the mismatch of the intuition and the tendency of the discrepancy in Theorem 2 is that, for its analysis, the meta-model error on the data collection meta-policy $\pi_\mathcal{D}$ (i.e., $\epsilon_m$) is used instead of that on the current meta-policy $\pi_\phi$. This ignorance of the meta-model error on the current meta-policy induces the pessimistic estimation of the discrepancy in the analysis (see the evaluation of "term B" and "term C" in the proof of Theorem 4 via the proof of Theorem 2 in the appendix for more details).

To estimate the discrepancy more tightly, as in Janner et al. (2019), we introduce the assumption that the meta-model error on the current meta-policy can be approximated by $\epsilon_\pi$ and $\epsilon_m$:

**Assumption 5.** *An approximated meta-model error on a current policy $\epsilon_{m'}$: $\epsilon_{m'}(\epsilon_\pi) \approx \epsilon_m + \epsilon_\pi \frac{d\epsilon_{m'}}{d\epsilon_\pi}$, where $\frac{d\epsilon_{m'}}{d\epsilon_\pi}$ is the local change of $\epsilon_{m'}$ with respect to $\epsilon_\pi$.*

To see the tendency of this approximated meta-model error, we plot the empirical value of $d\epsilon_{m'}/d\epsilon_\pi$, varying the size of training samples for the meta-model in Figure 10 in A.12. The figure shows that, as the training sample size increases, the value tends to gradually approach zero. This means that training meta-models with more samples provide better generalization on the nearby distributions.

Equipped with the approximated meta-model's error on the distribution of the current meta-policy $\pi_\phi$, we arrive at the following bound:

**Theorem 3.** *Let $\epsilon_{m'} \geq \max_t \mathbb{E}_{a \sim \pi_\phi, h \sim p} \left[ D_{TV} \left( p(h'|h, a) || p_\theta(h'|h, a) \right) \right]$,*

$$\mathbb{E}_{\pi_\phi, p}[R] \geq \mathbb{E}_{(a,h) \sim \mathcal{D}_{model}}[R] - r_{max} \left\{ \frac{1 + \gamma^2}{(1 - \gamma)^2} 2\epsilon_\pi + \frac{\gamma - k\gamma^k + (k - 1)\gamma^{k+1}}{(1 - \gamma)^2} (\epsilon_{m'} - \epsilon_\pi) \right.$$
$$\left. + \frac{\gamma^k - \gamma}{\gamma - 1} (\epsilon_{m'} - \epsilon_\pi) + \frac{\gamma^k}{1 - \gamma} (k + 1)(\epsilon_{m'} - \epsilon_\pi) \right\}. \quad (10)$$

*Proof.* By Lemma 1, our problem in POMDPs can be mapped into the problem in MDPs, and then Theorem 5 in A.7 can be applied to the problem. $\qquad \square$

Given that $\epsilon_m = \epsilon_{m'}$, it is obvious that the discrepancy in Theorem 3 is equal to or smaller than that in Theorem 2. In the discrepancy in Theorem 3, all terms except the first term become negative when $\epsilon_{m'} < \epsilon_\pi$. This implies that the optimal $k$ that minimizes the discrepancy can take on the value higher than 1 when the meta-model error is relatively small. The empirical trend of the discrepancy value (Figure 5b) supports it; when $\epsilon_{m'}$ is lower than 0.5 (i.e., $\epsilon_{m'} < \epsilon_\pi$), the discrepancy values decrease as the value of $k$ grows regardless of the value of $\gamma$. This result motivates us to set $k$ to the value higher than 1 in accordance with the meta-model error for reducing the discrepancy.

## A.7 THE DERIVATION OF THE RELATION OF THE RETURNS IN $k$-STEP BRANCHED ROLLOUTS ($k \geq 1$) IN MARKOV DECISION PROCESSES

In this section, we discuss the relation of the true returns and the model returns under the branched rollout in an MDP, which is defined by a tuple $\langle \mathcal{S}, \mathcal{A}, r, \gamma, p_{st} \rangle$. Here, $\mathcal{S}$ is the set of states, $\mathcal{A}$ is the set of actions, $p_{st} := p(s'|s, a)$ is the state transition probability for any $s', s \in \mathcal{S}$ and $a \in \mathcal{A}$, $r$ is the reward function and $\gamma \in [0, 1)$ is the discount factor. At time step $t$, the former two functions are used as $p(s_t|s_{t-1}, a_t)$, $r_t = r(s_t, a_t)$. The agent selects an action on the basis of a policy $\pi := p(a_{t+1}|s_t)$. We denote the data collection policy by $\pi_\mathcal{D}$ and the state visitation probability under $\pi_\mathcal{D}$ and $p(s'|s, a)$ by $p_{\pi_\mathcal{D}}(s_t)$. We also denote the predictive model for the next state by $p_\theta(s'|s, a)$. In addition, we define the upper bounds of the reward scale as $r_{max} > \max_{s,a}|r(s, a)|$. Note that, in this section, to discuss the MDP case, we are overriding the definition of the variables and functions that were defined for the POMDP case in the main body. In addition, for simplicity, we use the abbreviated style $\mathbb{E}_{\pi, p}[R]$ for the true return $\mathbb{E}_{a \sim \pi, s \sim p} \left[ R := \sum_{t=0}^{\infty} \gamma^t r_t \right]$.

Although the theoretical analysis on the relation of the returns in the MDP case is provided by Janner et al. (2019), in their analysis, a number of important premises are not properly taken into consideration. First, although they use the replay buffers for the branched rollout (i.e. datasets $\mathcal{D}_{env}$ and $\mathcal{D}_{model}$ in Algorithm 2 in Janner et al. (2019)), they do not take the use of the replay buffers into account in the their theoretical analysis. Furthermore, they calculate state-action visitation probabilities based solely on a single model-based rollout factor. In the branched rollout, state-action visitation probabilities (except for the one at $t = 0$) should be affected by multiple past model-based rollouts. For example, a state-action visitation probability at $t$ (s.t. $t > k$) is affected by the model-based rollout branched from real trajectories at $t - k$ and ones from $t - k + 1$ to $t - 1$ (in total, $k$ model-based rollouts). However, in their analysis (the proof of Lemma B.4 in Janner et al. (2019)), they calculate state-action visitation probabilities based solely on the model-based rollout. For example, in their analysis, it is assumed that a state-action visitation probability at $t$ (s.t. $t > k$) is affected only by the model-based rollout branched from real trajectories at $t - k$. These oversights of important premises in their analysis induce a large mismatch between those for their theorems and those made for the actual implementation of the branched rollout (i.e., Algorithm 2 in

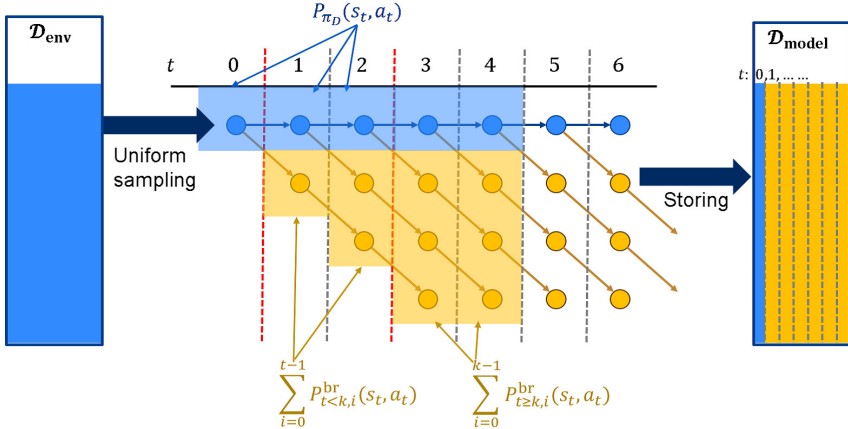

**Figure 6:** An example of branched rollouts with $k = 3$. Here, the blue dots represent trajectories contained in an environment dataset $\mathcal{D}_{\text{env}}$ and the yellow dots represent fictitious trajectories generated in accordance with a current policy $\pi$ under a predictive model $p_\theta$.

Janner et al. (2019)) [13]. Therefore, we decided to newly derive the theorems on the branched rollout, reflecting these premises more appropriately.

The outline of our branched rollout is shown in Figure 6. Here, we assume that the trajectories collected from the real environment are stored in a dataset $\mathcal{D}_{\text{env}}$. The trajectories stored in $\mathcal{D}_{\text{env}}$ can be seen as trajectories following the true dynamics $p(s'|s, a)$ and data collection policy (i.e., a mixture of previous policies used for data collection) $\pi_{\mathcal{D}}$. At each branched rollout, the trajectories in $D_{\text{env}}$ are uniformly sampled [14], and then starting from the sampled trajectories, $k$-step model-based rollouts in accordance with $\pi$ under $p_\theta$ is run. The fictitious trajectories generated by the branched rollout is stored in a model dataset $D_{\text{model}}$ [15]. This process more appropriately reflects the actual implementation of the branched rollout (i.e., lines 5–8 in Algorithm 2) in Janner et al. (2019) [16]. The performance of $\pi$ is evaluated as the expected return under the state-action visitation probability in $D_{\text{model}}$.

---

[13] The mismatch becomes larger especially when the model-rollout length $k$ is large because state-action visitation probabilities are affected by these rollouts more significantly with large $k$.

[14] Thus, the initial state probability for the rollout starting from the trajectories follows $p_{\pi_{\mathcal{D}}}(s)$

[15] Here, when the trajectories are stored in $D_{\text{model}}$, the states in the trajectories are augmented with time step information to deal with the state transition depending on the time step.

[16] Note that the extension of this process to the POMDP case is compatible with the implementation of the branched meta-rollout in our algorithm (lines 4–13 in Algorithm 2).

Formally, we define the return in the branched rollout $\mathbb{E}_{(a,s)\sim\mathcal{D}_{\text{model}}}[R]$ as:

$$
\begin{aligned}
\mathbb{E}_{(a,s)\sim\mathcal{D}_{\text{model}}}[R] \quad &:= \quad \sum_{s_0,a_0} p_{\pi_{\mathcal{D}}}(s_0,a_0)r(s_0,a_0) + \sum_{t=1}^{k-1}\sum_{s_t,a_t} \gamma^t p_{t<k}^{\text{br}}(s_t,a_t)r(s_t,a_t) \\
&\quad + \sum_{t=k}^{\infty}\sum_{s_t,a_t} \gamma^t p_{t\geq k}^{\text{br}}(s_t,a_t)r(s_t,a_t) \tag{11}
\end{aligned}
$$

$$
p_{t<k}^{\text{br}}(s_t,a_t) \quad := \quad \frac{1}{t}\sum_{i=0}^{t-1} p_{t<k,i}^{\text{br}}(s_t,a_t) \tag{12}
$$

$$
p_{t\geq k}^{\text{br}}(s_t,a_t) \quad := \quad \frac{1}{k}\sum_{i=0}^{k-1} p_{t\geq k,i}^{\text{br}}(s_t,a_t) \tag{13}
$$

$$
p_{t<k,i}^{\text{br}}(s_t,a_t) \quad := \quad \sum_{s_i,\dots,s_{t-1}}\sum_{a_i,\dots,a_{t-1}} p_{\pi_{\mathcal{D}}}(s_i)\Pi_{j=i}^{t-1}p_\theta(s_{j+1}|s_j,a_{j+1})\pi(a_{j+1}|s_j) \tag{14}
$$

$$
\begin{aligned}
p_{t\geq k,i}^{\text{br}}(s_t,a_t) \quad := \quad & \sum_{s_{t-k+i},\dots,s_{t-1}}\sum_{a_{t-k+i},\dots,a_{t-1}} \\
& p_{\pi_{\mathcal{D}}}(s_{t-k+i})\Pi_{j=t-k+i}^{t-1}p_\theta(s_{j+1}|s_j,a_{j+1})\pi(a_{j+1}|s_j) \tag{15}
\end{aligned}
$$

Here, $p_{t<k,i}^{\text{br}}(s_t,a_t)$ and $p_{t\geq k,i}^{\text{br}}(s_t,a_t)$ are the state-action visitation probabilities that the $i$-th yellow trajectories (node) from the bottom at each timestep $t$ in Figure 6 follow. In later discussions, for simplicity, we use the abbreviated style $\mathbb{E}_{\mathcal{D}_{\text{model}}}[R]$ for $\mathbb{E}_{(a,s)\sim\mathcal{D}_{\text{model}}}[R]$.

Before starting the derivation of theorems, we introduce a useful lemma.

**Lemma 2.** *Assume that the rollout process in which the policy and dynamics can be switched to other ones at time step $t_{sw}$. Letting two probabilities be $p_1$ and $p_2$, for $1 \leq t' \leq t_{sw}$, we assume that the dynamics distributions are bounded as $\epsilon_{m,pre} \geq \max_{t'} E_{s\sim p_1}[D_{TV}(p_1(s_{t'}|s_{t'-1},a_{t'})||p_2(s_{t'}|s_{t'-1},a_{t'}))]$. In addition, for $t_{sw} < t' \leq t$, we assume that the dynamics distributions are bounded as $\epsilon_{m,post} \geq \max_{t'} E_{s\sim p_1}[D_{TV}(p_1(s_{t'}|s_{t'-1},a_{t'})||p_2(s_{t'}|s_{t'-1},a_{t'}))]$. Likewise, the policy divergence is bounded by $\epsilon_{\pi,pre}$ and $\epsilon_{\pi,post}$. Then, the following inequation holds*

$$
\sum_{s_t,a_t}|p_1(s_t,a_t)-p_2(s_t,a_t)| \leq 2(t-t_{sw})(\epsilon_{m,post}+\epsilon_{\pi,post}) + 2t_{sw}(\epsilon_{m,pre}+\epsilon_{\pi,pre}) \tag{16}
$$

*Proof.* The proof is done in a similar manner to those of Lemma B.1 and B.2 in (Janner et al., 2019).

$$\sum_{s_t,a_t} |p_1(s_t,a_t) - p_2(s_t,a_t)|$$

$$= \sum_{s_t,a_t} |p_1(a_t)p_1(s_t|a_t) - p_2(a_t)p_2(s_t|a_t)|$$

$$= \sum_{s_t,a_t} |p_1(a_t)p_1(s_t|a_t) - p_1(a_t)p_2(s_t|a_t) + (p_1(a_t) - p_2(a_t))p_2(s_t|a_t)|$$

$$\leq \sum_{s_t,a_t} p_1(a_t)\,|p_1(s_t|a_t) - p_2(s_t|a_t)| + \sum_{a_t} |p_1(a_t) - p_2(a_t)|$$

$$\leq \sum_{s_t,a_t} p_1(a_t)\,|p_1(s_t|a_t) - p_2(s_t|a_t)| + \sum_{a_t,s_{t-1}} |p_1(a_t,s_{t-1}) - p_2(a_t,s_{t-1})|$$

$$= \sum_{s_t,a_t} p_1(a_t)\,|p_1(s_t|a_t) - p_2(s_t|a_t)|$$

$$+ \sum_{a_t,s_{t-1}} |p_1(s_{t-1})p_1(a_t|s_{t-1}) - p_1(s_{t-1})p_2(a_t|s_{t-1}) + (p_1(s_{t-1}) - p_2(s_{t-1}))p_2(a_t|s_{t-1})|$$

$$\leq \sum_{s_t,a_t} p_1(a_t)\,|p_1(s_t|a_t) - p_2(s_t|a_t)| + \sum_{a_t,s_{t-1}} p_1(s_{t-1})\,|p_1(a_t|s_{t-1}) - p_2(a_t|s_{t-1})|$$

$$+ \sum_{s_{t-1}} |p_1(s_{t-1}) - p_2(s_{t-1})|$$

$$\leq \sum_{s_t,a_t} p_1(a_t)\,|p_1(s_t|a_t) - p_2(s_t|a_t)| + \sum_{a_t,s_{t-1}} p_1(s_{t-1})\,|p_1(a_t|s_{t-1}) - p_2(a_t|s_{t-1})|$$

$$+ \sum_{s_{t-1},a_{t-1}} |p_1(s_{t-1},a_{t-1}) - p_2(s_{t-1},s_{t-1})|$$

$$\leq 2\epsilon_{m,\text{post}} + 2\epsilon_{\pi,\text{post}} + \sum_{s_{t-1},a_{t-1}} |p_1(s_{t-1},a_{t-1}) - p_2(s_{t-1},s_{t-1})|$$

$$\leq 2(t - t_{\text{sw}})(\epsilon_{m,\text{post}} + \epsilon_{\pi,\text{post}}) + \sum_{s_{t_{\text{sw}}},a_{t_{\text{sw}}}} |p_1(s_{t_{\text{sw}}},a_{t_{\text{sw}}}) - p_2(s_{t_{\text{sw}}},s_{t_{\text{sw}}})|$$

$$\leq 2(t - t_{\text{sw}})(\epsilon_{m,\text{post}} + \epsilon_{\pi,\text{post}}) + 2t_{\text{sw}}(\epsilon_{m,\text{pre}} + \epsilon_{\pi,\text{pre}})$$

$$(17)$$

$\square$

Now, we start the derivation of our bounds.

**Theorem 4.** *Under the $k$-step branched rollouts, using the bound of a model error under $\pi_\mathcal{D}$, $\epsilon_m = \max_t \mathbb{E}_{a\sim\pi_\mathcal{D},s\sim p,t}\left[D_{TV}(p(s'|s,a)||p_\theta(s'|s,a))\right]$ and the bound of the policy shift $\epsilon_\pi = \max_s D_{TV}(\pi||\pi_\mathcal{D})$, the following inequation holds,*

$$\mathbb{E}_{\pi,p}[R] \geq \mathbb{E}_{\mathcal{D}_{model}}[R] - r_{max}\left\{\frac{1+\gamma^2}{(1-\gamma)^2}2\epsilon_\pi + \frac{\gamma - k\gamma^k + (k-1)\gamma^{k+1}}{(1-\gamma)^2}(\epsilon_\pi + \epsilon_m)\right. \quad (18)$$

$$\left. + \frac{\gamma^k - \gamma}{\gamma - 1}(\epsilon_\pi + \epsilon_m) + \frac{\gamma^k}{1-\gamma}(k+1)(\epsilon_\pi + \epsilon_m)\right\}.$$

*Proof.*

$$
\left|\mathbb{E}_{\pi,p}\left[R\right] - \mathbb{E}_{\mathcal{D}_{\text{model}}}\left[R\right]\right| = \left|\begin{array}{l} \sum_{s_0,a_0}\left\{p_\pi(s_0,a_0) - p_{\pi_\mathcal{D}}(s_0,a_0)\right\}r(s_0,a_0) \\ + \sum_{t=1}^{k-1}\sum_{s_t,a_t}\gamma^t\left\{p_\pi(s_t,a_t) - p_{t<k}^{\text{br}}(s_t,a_t)\right\}r(s_t,a_t) \\ + \sum_{t=k}^{\infty}\sum_{s_t,a_t}\gamma^t\left\{p_\pi(s_t,a_t) - p_{t\geq k}^{\text{br}}(s_t,a_t)\right\}r(s_t,a_t)\end{array}\right|
$$

$$
\leq \left\{\begin{array}{l} \sum_{s_0,a_0}\left|p_\pi(s_0,a_0) - p_{\pi_\mathcal{D}}(s_0,a_0)\right|\left|r(s_0,a_0)\right| \\ + \sum_{t=1}^{k-1}\gamma^t\sum_{s_t,a_t}\left|p_\pi(s_t,a_t) - p_{t<k}^{\text{br}}(s_t,a_t)\right|\left|r(s_t,a_t)\right| \\ + \sum_{t=k}^{\infty}\gamma^t\sum_{s_t,a_t}\left|p_\pi(s_t,a_t) - p_{t\geq k}^{\text{br}}(s_t,a_t)\right|\left|r(s_t,a_t)\right|\end{array}\right\}
$$

$$
\leq \left\{\begin{array}{l} r_{\max}\underbrace{\sum_{s_0,a_0}\left|p_\pi(s_0,a_0) - p_{\pi_\mathcal{D}}(s_0,a_0)\right|}_{\text{term A}} \\[2em] +r_{\max}\sum_{t=1}^{k-1}\gamma^t\underbrace{\sum_{s_t,a_t}\left|p_\pi(s_t,a_t) - p_{t<k}^{\text{br}}(s_t,a_t)\right|}_{\text{term B}} \\[2em] +r_{\max}\sum_{t=k}^{\infty}\gamma^t\underbrace{\sum_{s_t,a_t}\left|p_\pi(s_t,a_t) - p_{t\geq k}^{\text{br}}(s_t,a_t)\right|}_{\text{term C}}\end{array}\right\} \tag{19}
$$

For **term A**, we can bound the value in similar manner to the derivation of Lemma 2.

$$
\sum_{s_0,a_0}\left|p_\pi(s_0,a_0) - p_{\pi_\mathcal{D}}(s_0,a_0)\right| = \sum_{s_0,a_0}\left|p_\pi(a_0)p(s_0) - p_{\pi_\mathcal{D}}(a_0)p(s_0)\right|
$$

$$
= \sum_{s_0,a_0}\left|p_\pi(a_0)p(s_0) - p_\pi(a_0)p(s_0) + \left(p_\pi(a_0) - p_{\pi_\mathcal{D}}(a_0)\right)p(s_0)\right|
$$

$$
\leq \underbrace{\sum_{s_0,a_0}p_\pi(a_0)\left|p(s_0) - p(s_0)\right|}_{=0} + \underbrace{\sum_{a_0}\left|p_\pi(a_0) - p_{\pi_\mathcal{D}}(a_0)\right|}_{\leq 2\epsilon_\pi}
$$

$$
\leq 2\epsilon_\pi \tag{20}
$$

For **term B**, we can apply Lemma 2 to bound the value, but it requires the bounded model error under the current policy $\pi$. Thus, we need to decompose the distance into two by adding and subtracting $p_{\pi_\mathcal{D}}$:

$$
\sum_{s_t,a_t}\left|p_\pi(s_t,a_t) - p_{\text{br},t<k}(s_t,a_t)\right| = \sum_{s_t,a_t}\left|\begin{array}{l}p_\pi(s_t,a_t) - p_{\pi_\mathcal{D}}(s_t,a_t) \\ +p_{\pi_\mathcal{D}}(s_t,a_t) - p_{t<k}^{\text{br}}(s_t,a_t)\end{array}\right|
$$

$$
\leq \underbrace{\sum_{s_t,a_t}\left|p_\pi(s_t,a_t) - p_{\pi_\mathcal{D}}(s_t,a_t)\right|}_{\leq 2t\epsilon_\pi}
$$

$$
+ \sum_{s_t,a_t}\left|p_{\pi_\mathcal{D}}(s_t,a_t) - p_{t<k}^{\text{br}}(s_t,a_t)\right| \tag{21}
$$

$$
\sum_{s_t,a_t}\left|p_{\pi_\mathcal{D}}(s_t,a_t) - p_{t<k}^{\text{br}}(s_t,a_t)\right| = \sum_{s_t,a_t}\left|\frac{1}{t}\sum_{i=0}^{t-1}p_{\pi_\mathcal{D}}(s_t,a_t) - \frac{1}{t}\sum_{i=0}^{t-1}p_{t<k,i}^{\text{br}}(s_t,a_t)\right|
$$

$$
\leq \frac{1}{t}\sum_{i=0}^{t-1}\sum_{s_t,a_t}\left|p_{\pi_\mathcal{D}}(s_t,a_t) - p_{t<k,i}^{\text{br}}(s_t,a_t)\right|
$$

$$
\overset{(A)}{\leq} \frac{1}{t}\sum_{i=0}^{t-1}\left\{2\left(t-i\right)\cdot\left(\epsilon_\pi + \epsilon_m\right)\right\}
$$

$$
= \frac{1}{t}\left\{t^2(\epsilon_\pi + \epsilon_m) + t(\epsilon_\pi + \epsilon_m)\right\} \tag{22}
$$

For (A), we apply Lemma 2 with setting $\epsilon_{m,\text{post}} = \epsilon_m$ and $\epsilon_{\pi,\text{post}} = \epsilon_\pi$ for the rollout following $\pi$ and $p_\theta(s'|s)$, and $\epsilon_{m,\text{pre}} = 0$ and $\epsilon_{\pi,\text{pre}} = 0$ for the rollout following $\pi_\mathcal{D}$ and $p(s'|s)$, respectively. To recap **term B**, the following equation holds:

$$\sum_{s_t,a_t} |p_\pi(s_t,a_t) - p_{\text{br},t<k}(s_t,a_t)| \leq 2t\epsilon_\pi + t(\epsilon_\pi + \epsilon_m) + (\epsilon_\pi + \epsilon_m) \tag{23}$$

For **term C**, we can derive the bound in a similar manner to the term B case:

$$\sum_{s_t,a_t} |p_\pi(s_t,a_t) - p_{\text{br},t\geq k}(s_t,a_t)| = \sum_{s_t,a_t} \left| \begin{array}{c} p_\pi(s_t,a_t) - p_{\pi_\mathcal{D}}(s_t,a_t) \\ +p_{\pi_\mathcal{D}}(s_t,a_t) - p_{t\geq k}^{\text{br}}(s_t,a_t) \end{array} \right|$$

$$\leq \underbrace{\sum_{s_t,a_t} |p_\pi(s_t,a_t) - p_{\pi_\mathcal{D}}(s_t,a_t)|}_{\leq 2t\epsilon_\pi}$$

$$+ \sum_{s_t,a_t} \left| p_{\pi_\mathcal{D}}(s_t,a_t) - p_{t\geq k}^{\text{br}}(s_t,a_t) \right| \tag{24}$$

$$\sum_{s_t,a_t} \left| p_{\pi_\mathcal{D}}(s_t,a_t) - p_{t\geq k}^{\text{br}}(s_t,a_t) \right| = \sum_{s_t,a_t} \left| \tfrac{1}{k}\sum_{i=0}^{k-1} p_{\pi_\mathcal{D}}(s_t,a_t) - \tfrac{1}{k}\sum_{i=0}^{k-1} p_{t\geq k,i}^{\text{br}}(s_t,a_t) \right|$$

$$\leq \frac{1}{k}\sum_{i=0}^{k-1}\sum_{s_t,a_t} \left| p_{\pi_\mathcal{D}}(s_t,a_t) - p_{t\geq k,i}^{\text{br}}(s_t,a_t) \right|$$

$$\leq \frac{1}{k}\sum_{i=0}^{k-1} \{2(k-i)\cdot(\epsilon_\pi + \epsilon_m)\}$$

$$= \frac{1}{k}\{k^2(\epsilon_\pi + \epsilon_m) + k(\epsilon_\pi + \epsilon_m)\} \tag{25}$$

To recap **term C**, the following equation holds:

$$\sum_{s_t,a_t} \left| p_\pi(s_t,a_t) - p_{t\geq k}^{\text{br}}(s_t,a_t) \right| \leq 2t\epsilon_\pi + k(\epsilon_\pi + \epsilon_m) + (\epsilon_\pi + \epsilon_m) \tag{26}$$

By substituting Eqs. 20, 23, and 26, into Eq. 19, we obtain the result:

$$
\big|\mathbb{E}_{\pi,p}\left[R\right] - \mathbb{E}_{\mathcal{D}_{\mathrm{model}}}\left[R\right]\big| \leq
\left\{
\begin{array}{c}
r_{\max} 2\epsilon_\pi \\
+r_{\max}\sum_{t=1}^{k-1}\gamma^t\left\{2t\epsilon_\pi + t(\epsilon_\pi+\epsilon_m) + (\epsilon_\pi+\epsilon_m)\right\} \\
+r_{\max}\sum_{t=k}^{\infty}\gamma^t\left\{2t\epsilon_\pi + k(\epsilon_\pi+\epsilon_m) + (\epsilon_\pi+\epsilon_m)\right\}
\end{array}
\right\}
$$

$$
= r_{\max}\left\{
\begin{array}{c}
2\epsilon_\pi + \frac{1-k\gamma^{(k-1)}+(k-1)\gamma^k}{(1-\gamma)^2}\gamma\left(3\epsilon_\pi+\epsilon_m\right) + \frac{\gamma^k-\gamma}{\gamma-1}(\epsilon_\pi+\epsilon_m) \\
+\sum_{t=k}^{\infty}\gamma^t\left\{2t\epsilon_\pi + k(\epsilon_\pi+\epsilon_m) + (\epsilon_\pi+\epsilon_m)\right\}
\end{array}
\right\}
$$

$$
= r_{\max}\left\{
\begin{array}{c}
2\epsilon_\pi + \frac{1-k\gamma^{(k-1)}+(k-1)\gamma^k}{(1-\gamma)^2}\gamma\left(3\epsilon_\pi+\epsilon_m\right) + \frac{\gamma^k-\gamma}{\gamma-1}(\epsilon_\pi+\epsilon_m) \\
+\sum_{t=1}^{\infty}\gamma^t\left\{2t\epsilon_\pi + k(\epsilon_\pi+\epsilon_m) + (\epsilon_\pi+\epsilon_m)\right\} \\
-\sum_{t=1}^{k-1}\gamma^t\left\{2t\epsilon_\pi + k(\epsilon_\pi+\epsilon_m) + (\epsilon_\pi+\epsilon_m)\right\}
\end{array}
\right\}
$$

$$
= r_{\max}\left\{
\begin{array}{c}
2\epsilon_\pi + \frac{1-k\gamma^{(k-1)}+(k-1)\gamma^k}{(1-\gamma)^2}\gamma\left(3\epsilon_\pi+\epsilon_m\right) + \frac{\gamma^k-\gamma}{\gamma-1}(\epsilon_\pi+\epsilon_m) \\
+\frac{2}{(1-\gamma)^2}\gamma\epsilon_\pi + \frac{\gamma}{1-\gamma}\left\{k(\epsilon_\pi+\epsilon_m) + (\epsilon_\pi+\epsilon_m)\right\} \\
-\sum_{t=1}^{k-1}\gamma^t\left\{2t\epsilon_\pi + k(\epsilon_\pi+\epsilon_m) + (\epsilon_\pi+\epsilon_m)\right\}
\end{array}
\right\}
$$

$$
= r_{\max}\left\{
\begin{array}{c}
2\epsilon_\pi + \frac{1-k\gamma^{(k-1)}+(k-1)\gamma^k}{(1-\gamma)^2}\gamma\left(3\epsilon_\pi+\epsilon_m\right) + \frac{\gamma^k-\gamma}{\gamma-1}(\epsilon_\pi+\epsilon_m) \\
+\frac{2}{(1-\gamma)^2}\gamma\epsilon_\pi + \frac{\gamma}{1-\gamma}\left\{k(\epsilon_\pi+\epsilon_m) + (\epsilon_\pi+\epsilon_m)\right\} \\
-\frac{1-k\gamma^{(k-1)}+(k-1)\gamma^k}{(1-\gamma)^2}2\gamma\epsilon_\pi \\
-\frac{\gamma^k-\gamma}{\gamma-1}\left\{k(\epsilon_\pi+\epsilon_m) + (\epsilon_\pi+\epsilon_m)\right\}
\end{array}
\right\}
$$

$$
= r_{\max}\left\{
\begin{array}{c}
\frac{1+\gamma^2}{(1-\gamma)^2}2\epsilon_\pi + \frac{\gamma-k\gamma^k+(k-1)\gamma^{k+1}}{(1-\gamma)^2}(\epsilon_\pi+\epsilon_m) \\
+\frac{\gamma^k-\gamma}{\gamma-1}(\epsilon_\pi+\epsilon_m) + \left(\frac{\gamma}{1-\gamma}-\frac{\gamma^k-\gamma}{\gamma-1}\right)(k+1)(\epsilon_\pi+\epsilon_m)
\end{array}
\right\}
$$

$$
= r_{\max}\left\{
\begin{array}{c}
\frac{1+\gamma^2}{(1-\gamma)^2}2\epsilon_\pi + \frac{\gamma-k\gamma^k+(k-1)\gamma^{k+1}}{(1-\gamma)^2}(\epsilon_\pi+\epsilon_m) \\
+\frac{\gamma^k-\gamma}{\gamma-1}(\epsilon_\pi+\epsilon_m) + \frac{\gamma^k}{1-\gamma}(k+1)(\epsilon_\pi+\epsilon_m)
\end{array}
\right\}
\quad (27)
$$

$\square$

**Theorem 5.** *Let* $\epsilon_{m'} \geq \max_t \mathbb{E}_{a\sim\pi,s\sim p}\left[D_{TV}\left(p(s'|s,a)||p_\theta(s'|s,a)\right)\right]$,

$$
\mathbb{E}_{\pi,p}\left[R\right] \geq \mathbb{E}_{\mathcal{D}_{model}}\left[R\right] - r_{max}\left\{\frac{1+\gamma^2}{(1-\gamma)^2}2\epsilon_\pi + \frac{1-k\gamma^{(k-1)}+(k-1)\gamma^k}{(1-\gamma)^2}\gamma(\epsilon_{m'}-\epsilon_\pi)\right.
$$
$$
\left. + \frac{\gamma^k-\gamma}{\gamma-1}(\epsilon_{m'}-\epsilon_\pi) + \frac{\gamma^k}{1-\gamma}(k+1)(\epsilon_{m'}-\epsilon_\pi)\right\}. \quad (28)
$$

*Proof.* The derivation of this theorem is basically the same as that in the Theorem 4 case except for the way of evaluation of the bound of terms B and C.

For **term B**, we can apply Lemma 2 to bound the value:

$$
\sum_{s_t,a_t}\left|p_\pi(s_t,a_t) - p_{t<k}^{\mathrm{br}}(s_t,a_t)\right| = \sum_{s_t,a_t}\left|\frac{1}{t}\sum_{i=0}^{t-1}p_{\pi_\mathcal{D}}(s_t,a_t) - \frac{1}{t}\sum_{i=0}^{t-1}p_{t<k,i}^{\mathrm{br}}(s_t,a_t)\right|
$$

$$
\leq \frac{1}{t}\sum_{i=0}^{t-1}\sum_{s_t,a_t}\left|p_\pi(s_t,a_t) - p_{t<k,i}^{\mathrm{br}}(s_t,a_t)\right|
$$

$$
\overset{(D)}{\leq} \frac{1}{t}\sum_{i=0}^{t-1}\left\{(t-i)2\epsilon_{m'} + i2\epsilon_\pi\right\}
$$

$$
= \frac{1}{t}\left\{2t^2\epsilon_{m'} - (t-1)t\epsilon_{m'} + (t-1)t\epsilon_\pi\right\}
$$

$$
= \frac{1}{t}\left\{2t^2\epsilon_{m'} - t^2\epsilon_{m'} + t\epsilon_{m'} + t^2\epsilon_\pi - t\epsilon_\pi\right\}
$$

$$
= \frac{1}{t}\left\{t^2(\epsilon_{m'}+\epsilon_\pi) + t(\epsilon_{m'}-\epsilon_\pi)\right\}
$$

$$
= t(\epsilon_{m'}+\epsilon_\pi) + (\epsilon_{m'}-\epsilon_\pi) \quad (29)
$$

For (D), we apply Lemma 2 with setting $\epsilon_{m,\text{post}} = \epsilon_{m'}$ and $\epsilon_{\pi,\text{post}} = 0$ for the rollout following $\pi$ and $p_\theta(s'|s)$, and $\epsilon_{m,\text{pre}} = 0$ and $\epsilon_{\pi,\text{pre}} = \epsilon_\pi$ for the rollout following $\pi_\mathcal{D}$ and $p(s'|s)$.

For **term C**, we can derive the bound in a similar manner to the case of term B:

$$
\begin{aligned}
\sum_{s_t,a_t} \left| p_\pi(s_t,a_t) - p_{t\geq k}^{\text{br}}(s_t,a_t) \right| &= \sum_{s_t,a_t} \left| \tfrac{1}{k}\sum_{i=0}^{k-1} p_{\pi_\mathcal{D}}(s_t,a_t) - \tfrac{1}{k}\sum_{i=0}^{k-1} p_{t\geq k,i}^{\text{br}}(s_t,a_t) \right| \\
&\leq \frac{1}{k}\sum_{i=0}^{k-1}\sum_{s_t,a_t} \left| p_\pi(s_t,a_t) - p_{t\geq k,i}^{\text{br}}(s_t,a_t) \right| \\
&\leq \frac{1}{k}\sum_{i=0}^{k-1}\left\{(k-i)2\epsilon_{m'} + (t-k+i)2\epsilon_\pi\right\} \\
&= \frac{1}{k}\left\{2k^2\epsilon_{m'} - (k-1)k\epsilon_{m'} + 2tk\epsilon_\pi - 2k^2\epsilon_\pi + (k-1)k\epsilon_\pi\right\} \\
&= \frac{1}{k}\left\{2kt\epsilon_\pi + 2(\epsilon_{m'}-\epsilon_\pi)k^2 - (k-1)k(\epsilon_{m'}-\epsilon_\pi)\right\} \\
&= \frac{1}{k}\left\{2kt\epsilon_\pi + (\epsilon_{m'}-\epsilon_\pi)k^2 + k(\epsilon_{m'}-\epsilon_\pi)\right\} \qquad (30)
\end{aligned}
$$

By substituting Eqs. 20, 29, and 30, into Eq. 19, we obtain the result:

$$
\begin{aligned}
\left|\mathbb{E}_{\pi,p}[R] - \mathbb{E}_{\mathcal{D}_{\text{model}}}[R]\right| &\leq r_{\max}\left\{\begin{array}{l} 2\epsilon_\pi + \sum_{t=1}^{k-1}\gamma^t\left\{t(\epsilon_{m'}+\epsilon_\pi)+(\epsilon_{m'}-\epsilon_\pi)\right\} \\ +\sum_{t=k}^{\infty}\gamma^t\left\{\tfrac{1}{k}\left\{2kt\epsilon_\pi+(\epsilon_{m'}-\epsilon_\pi)k^2+k(\epsilon_{m'}-\epsilon_\pi)\right\}\right\}\end{array}\right\} \\
&= r_{\max}\left\{\begin{array}{l} 2\epsilon_\pi + \frac{1-k\gamma^{(k-1)}+(k-1)\gamma^k}{(1-\gamma)^2}\gamma(\epsilon_{m'}+\epsilon_\pi) + \frac{\gamma^k-\gamma}{\gamma-1}(\epsilon_{m'}-\epsilon_\pi) \\ +\sum_{t=k}^{\infty}\gamma^t\tfrac{1}{k}\left\{2kt\epsilon_\pi+(\epsilon_{m'}-\epsilon_\pi)k^2+k(\epsilon_{m'}-\epsilon_\pi)\right\}\end{array}\right\} \\
&= r_{\max}\left\{\begin{array}{l} 2\epsilon_\pi + \frac{1-k\gamma^{(k-1)}+(k-1)\gamma^k}{(1-\gamma)^2}\gamma(\epsilon_{m'}+\epsilon_\pi) + \frac{\gamma^k-\gamma}{\gamma-1}(\epsilon_{m'}-\epsilon_\pi) \\ +\sum_{t=1}^{\infty}\gamma^t\tfrac{1}{k}\left\{2kt\epsilon_\pi+(\epsilon_{m'}-\epsilon_\pi)k^2+k(\epsilon_{m'}-\epsilon_\pi)\right\} \\ -\sum_{t=1}^{k-1}\gamma^t\tfrac{1}{k}\left\{2kt\epsilon_\pi+(\epsilon_{m'}-\epsilon_\pi)k^2+k(\epsilon_{m'}-\epsilon_\pi)\right\}\end{array}\right\} \\
&= r_{\max}\left\{\begin{array}{l} 2\epsilon_\pi + \frac{1-k\gamma^{(k-1)}+(k-1)\gamma^k}{(1-\gamma)^2}\gamma(\epsilon_{m'}+\epsilon_\pi) + \frac{\gamma^k-\gamma}{\gamma-1}(\epsilon_{m'}-\epsilon_\pi) \\ +\frac{1}{(1-\gamma)^2}2\gamma\epsilon_\pi + \frac{\gamma}{1-\gamma}\left\{(\epsilon_{m'}-\epsilon_\pi)k+(\epsilon_{m'}-\epsilon_\pi)\right\} \\ \quad - \frac{1-k\gamma^{(k-1)}+(k-1)\gamma^k}{(1-\gamma)^2}2\gamma\epsilon_\pi \\ \quad -\frac{\gamma^k-\gamma}{\gamma-1}\left\{(\epsilon_{m'}-\epsilon_\pi)k+(\epsilon_{m'}-\epsilon_\pi)\right\}\end{array}\right\} \\
&= r_{\max}\left\{\begin{array}{l} \frac{1+\gamma^2}{(1-\gamma)^2}2\epsilon_\pi + \frac{\gamma-k\gamma^k+(k-1)\gamma^{k+1}}{(1-\gamma)^2}(\epsilon_{m'}-\epsilon_\pi) \\ +\frac{\gamma^k-\gamma}{\gamma-1}(\epsilon_{m'}-\epsilon_\pi) + \left(\frac{\gamma}{1-\gamma}-\frac{\gamma^k-\gamma}{\gamma-1}\right)(k+1)(\epsilon_{m'}-\epsilon_\pi)\end{array}\right\} \\
&= r_{\max}\left\{\begin{array}{l} \frac{1+\gamma^2}{(1-\gamma)^2}2\epsilon_\pi + \frac{\gamma-k\gamma^k+(k-1)\gamma^{k+1}}{(1-\gamma)^2}(\epsilon_{m'}-\epsilon_\pi) \\ +\frac{\gamma^k-\gamma}{\gamma-1}(\epsilon_{m'}-\epsilon_\pi) + \frac{\gamma^k}{1-\gamma}(k+1)(\epsilon_{m'}-\epsilon_\pi)\end{array}\right\} \qquad (31)
\end{aligned}
$$

$\square$

## A.8 The relation of returns in the case where a reward prediction is inaccurate

In Section 5, we discuss the relation between the true returns and the returns estimated on the meta-model under the assumption that the reward prediction error is zero. The theoretical result under this assumption is still useful because there are many cases where the true reward function is given and the reward prediction is not required. However, a number of readers still may want to know what the relation of the returns is under the assumption that the reward prediction is inaccurate. In this section, we provide the relation of the returns under inaccurate reward prediction in the MDP case [17].

---

[17]Here, we do not discuss the theorems in the POMDP case because those in the MDP case can be easily extended into the POMDP case by utilizing Lemma 1.

We start our discussion by defining the bound of the reward prediction error $\epsilon_r$:

**Definition 4.** $\epsilon_r := \max_t \mathbb{E}_{(a_t, s_t) \sim \mathcal{D}_{model}} [|r(s_t, a_t) - r_\theta(s_t, a_t)|]$, where $r_\theta(s_t, a_t) := \mathbb{E}_{r_t \sim p_\theta}[r_t | a_t, s_t]$.

We also define the return on the branched rollout with inaccurate reward prediction.

$$
\mathbb{E}_{\mathcal{D}_{\text{model}}}[\hat{R}] \quad := \quad \sum_{s_0, a_0} p_{\pi_\mathcal{D}}(s_0, a_0) r_\theta(s_0, a_0) + \sum_{t=1}^{k-1} \sum_{s_t, a_t} \gamma^t p_{t<k}^{\text{br}}(s_t, a_t) r_\theta(s_t, a_t)
$$
$$
+ \sum_{t=k}^{\infty} \sum_{s_t, a_t} \gamma^t p_{t \geq k}^{\text{br}}(s_t, a_t) r_\theta(s_t, a_t) \tag{32}
$$

Now, we provide the relation between the returns under inaccurate reward prediction.

**Theorem 6** (Extension of Theorem 4 into the case where reward prediction is inaccurate)**.** *Under the $k$ steps branched rollouts, using the bound of a model error under $\pi_\mathcal{D}$, $\epsilon_m = \max_t \mathbb{E}_{a \sim \pi_\mathcal{D}, s \sim p, t} [D_{TV}(p(s'|s, a) || p_\theta(s'|s, a))]$, the bound of the policy shift $\epsilon_\pi = \max_s D_{TV}(\pi || \pi_\mathcal{D})$ and the bound of the reward prediction error $\epsilon_r = \max_t \mathbb{E}_{(a_t, s_t) \sim \mathcal{D}_{model}} [|r(s_t, a_t) - r_\theta(s_t, a_t)|]$, the following inequation holds,*

$$
\mathbb{E}_{\pi, p}[R] \geq \mathbb{E}_{\mathcal{D}_{model}}[\hat{R}] - r_{max} \left\{ \frac{1 + \gamma^2}{(1 - \gamma)^2} 2\epsilon_\pi + \frac{\gamma - k\gamma^k + (k-1)\gamma^{k+1}}{(1 - \gamma)^2} (\epsilon_\pi + \epsilon_m) \right.
$$
$$
\left. + \frac{\gamma^k - \gamma}{\gamma - 1} (\epsilon_\pi + \epsilon_m) + \frac{\gamma^k}{1 - \gamma} (k+1)(\epsilon_\pi + \epsilon_m) \right\} - \frac{\gamma}{1 - \gamma} \epsilon_r. \tag{33}
$$

*Proof.*

$$
\left| \mathbb{E}_{\pi, p}[R] - \mathbb{E}_{\mathcal{D}_{\text{model}}}[\hat{R}] \right| = \left| \mathbb{E}_{\pi, p}[R] - \mathbb{E}_{\mathcal{D}_{\text{model}}}[R] + \mathbb{E}_{\mathcal{D}_{\text{model}}}[R] - \mathbb{E}_{\mathcal{D}_{\text{model}}}[\hat{R}] \right| \tag{34}
$$
$$
\leq \left| \mathbb{E}_{\pi, p}[R] - \mathbb{E}_{\mathcal{D}_{\text{model}}}[R] \right| + \left| \mathbb{E}_{\mathcal{D}_{\text{model}}}[R] - \mathbb{E}_{\mathcal{D}_{\text{model}}}[\hat{R}] \right|
$$

$$
\left| \mathbb{E}_{\mathcal{D}_{\text{model}}}[R] - \mathbb{E}_{\mathcal{D}_{\text{model}}}[\hat{R}] \right| = \left| \begin{array}{l} \sum_{s_0, a_0} p_{\pi_\mathcal{D}}(s_0, a_0) \{r(s_0, a_0) - r_\theta(s_0, a_0)\} \\ + \sum_{t=1}^{k-1} \sum_{s_t, a_t} \gamma^t p_{t<k}^{\text{br}}(s_t, a_t) \{r(s_t, a_t) - r_\theta(s_t, a_t)\} \\ + \sum_{t=k}^{\infty} \sum_{s_t, a_t} \gamma^t p_{t \geq k}^{\text{br}}(s_t, a_t) \{r(s_t, a_t) - r_\theta(s_t, a_t)\} \end{array} \right|
$$
$$
\leq \sum_{s_0, a_0} p_{\pi_\mathcal{D}}(s_0, a_0) |r(s_0, a_0) - r_\theta(s_0, a_0)|
$$
$$
+ \sum_{t=1}^{k-1} \sum_{s_t, a_t} \gamma^t p_{t<k}^{\text{br}}(s_t, a_t) |r(s_t, a_t) - r_\theta(s_t, a_t)| \tag{35}
$$
$$
+ \sum_{t=k}^{\infty} \sum_{s_t, a_t} \gamma^t p_{t \geq k}^{\text{br}}(s_t, a_t) |r(s_t, a_t) - r_\theta(s_t, a_t)|
$$
$$
\leq \sum_{t=0}^{\infty} \gamma^t \epsilon_r
$$
$$
= \frac{\gamma}{1 - \gamma} \epsilon_r
$$

By substituting Eqs. 27 and 35 into Eq. 34, we obtain the result:

$$
\left| \mathbb{E}_{\pi, p}[R] - \mathbb{E}_{\mathcal{D}_{\text{model}}}[\hat{R}] \right| \leq r_{\max} \left\{ \begin{array}{l} \frac{1 + \gamma^2}{(1 - \gamma)^2} 2\epsilon_\pi + \frac{\gamma - k\gamma^k + (k-1)\gamma^{k+1}}{(1 - \gamma)^2} (\epsilon_\pi + \epsilon_m) \\ + \frac{\gamma^k - \gamma}{\gamma - 1} (\epsilon_\pi + \epsilon_m) + \frac{\gamma^k}{1 - \gamma} (k+1)(\epsilon_\pi + \epsilon_m) \end{array} \right\} + \frac{\gamma}{1 - \gamma} \epsilon_r \tag{36}
$$

$\square$

**Theorem 7** (Extension of Theorem 5 into the case where reward prediction is inaccurate). *Let* $\epsilon_{m'} \geq \max_t E_{a \sim \pi, s \sim p} [D_{TV}(p(s'|s,a)||p_\theta(s'|s,a))]$ *and* $\epsilon_r = \max_t \mathbb{E}_{(a_t, s_t) \sim \mathcal{D}_{model}} [|r(s_t, a_t) - r_\theta(s_t, a_t)|]$,

$$\mathbb{E}_{\pi, p}[R] \geq \mathbb{E}_{\mathcal{D}_{model}}[\hat{R}] - r_{max} \left\{ \frac{1 + \gamma^2}{(1-\gamma)^2} 2\epsilon_\pi + \frac{\gamma - k\gamma^k + (k-1)\gamma^{k+1}}{(1-\gamma)^2} (\epsilon_{m'} - \epsilon_\pi) \right.$$
$$\left. + \frac{\gamma^k - \gamma}{\gamma - 1} (\epsilon_{m'} - \epsilon_\pi) + \frac{\gamma^k}{1-\gamma} (k+1)(\epsilon_{m'} - \epsilon_\pi) \right\} - \frac{\gamma}{1-\gamma} \epsilon_r. \quad (37)$$

*Proof.* Similar to the derivation of Theorem 6, we obtain the result by substituting Eq. 31 and 35 into Eq. 34. ☐

### A.9    PEARL in Sections 6 and 7

The PEARL algorithm used in Sections 6 and 7 refers to "PEARL with RNN-traj" in (Rakelly et al., 2019). The comparison result with other types of PEARL (i.e., vanilla PEARL and PEARL with RNN-tran) in (Rakelly et al., 2019) and M3PO are shown in Figure 7. The figure indicates that M3PO achieves better sample efficiency than them.

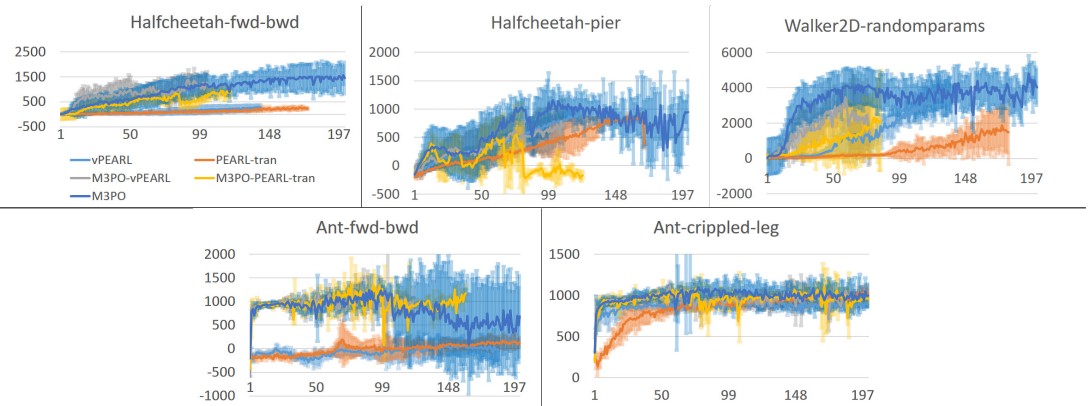

**Figure 7:** The learning curve of vanilla PEARL (vPEARL), PEARL with RNN-traj (PEARL-tran), M3PO, and its variants in which the vPEARL and PEARL-tran are used for meta-policies. In each figure, the vertical axis represents expected returns and the horizontal axis represents the number of training samples (x1000). The meta-policy and meta-model (and a predictive model) were fixed and their expected returns were evaluated on 50 episodes at every 1000 training samples for the other methods. In each episode, the task was initialized and changed randomly. Each method was evaluated in at least three trials, and the expected return on the 50 episodes was further averaged over the trials. The averaged expected returns and their standard deviations are plotted in the figures.

### A.10    Baseline methods for our experiment

**PEARL:** The model-free meta-RL method proposed in Rakelly et al. (2019). This is an off-policy method and implemented by extending Soft Actor-Critic (Haarnoja et al., 2018). By leveraging experience replay, this method shows high sample efficiency. We reimplemented the PEARL algorithm on TensorFlow, referring to the original implementation on PyTorch (`https://github.com/katerakelly/oyster`).
**Learning to adapt (L2A):** The model-based meta-RL proposed in Nagabandi et al. (2019a). In this method, the meta-model is implemented with MAML (Finn et al., 2017) and the optimal action is found by the model predictive path integral control (Williams et al., 2015) on the full meta–model based rollouts. We adapt the following implementation of L2A to our experiment: `https://github.com/iclavera/learning_to_adapt`

## A.11 Environments for our experiment

For our experiment in Section 7, we prepare simulated robot environments using the MuJoCo physics engine (Todorov et al., 2012) (Figure 8):

**Halfcheetah-fwd-bwd:** In this environment, meta-policies are used to control the half-cheetah, which is a planar biped robot with eight rigid links, including two legs and a torso, along with six actuated joints. Here, the half-cheetah's moving direction is randomly selected from "forward" and "backward" around every 15 seconds (in simulation time). If the half-cheetah moves in the correct direction, a positive reward is fed to the half-cheetah in accordance with the magnitude of movement, otherwise, a negative reward is fed.

**Halfcheetah-pier:** In this environment, the half-cheetah runs over a series of blocks that are floating on water. Each block moves up and down when stepped on, and the changes in the dynamics are rapidly changing due to each block having different damping and friction properties. These properties are randomly determined at the beginning of each episode.

**Ant-fwd-bwd:** Same as Halfcheetah-fwd-bwd except that the meta-policies are used for controlling the ant, which is a quadruped robot with nine rigid links, including four legs and a torso, along with eight actuated joints.

**Ant-crippled-leg:** In this environment, we randomly sample a leg on the ant to cripple. The crippling of the leg causes unexpected and drastic changes to the underlying dynamics. One of the four legs is randomly crippled every 15 seconds.

**Walker2D-randomparams:** In this environment, the meta-policies are used to control the walker, which is a planar biped robot consisting of seven links, including two legs and a torso, along with six actuated joints. The walker's torso mass and ground friction is randomly determined every 15 seconds.

**Humanoid-direc:** In this environment, the meta-policies are used to control the humanoid, which is a biped robot with 13 rigid links, including two legs, two arms and a torso, along with 17 actuated joints. In this task, the humanoid moving direction is randomly selected from two different directions around every 15 seconds. If the humanoid moves in the correct direction, a positive reward is fed to the humanoid in accordance with the magnitude of its movement, otherwise, a negative reward is fed.

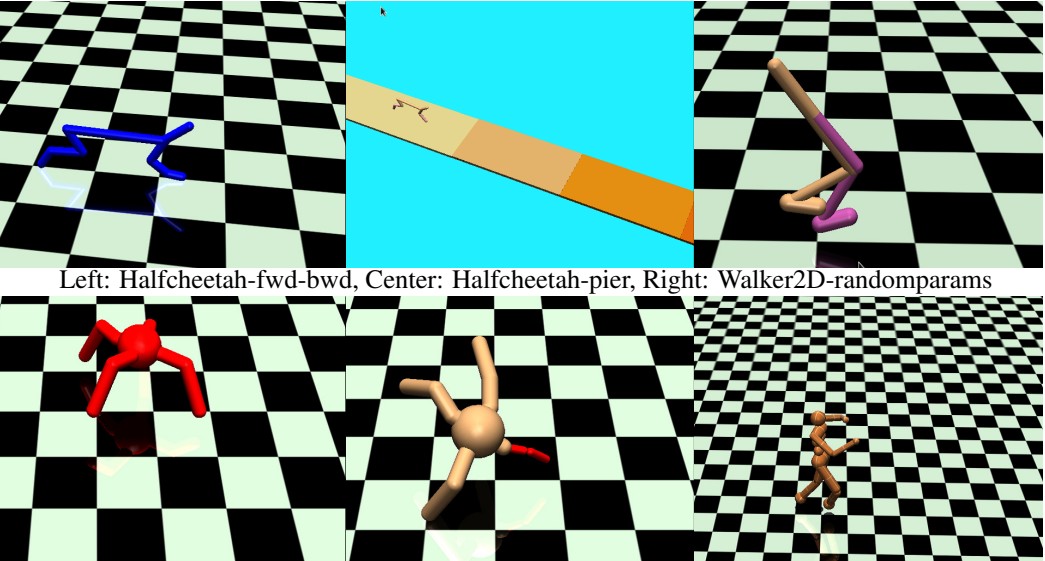

Left: Halfcheetah-fwd-bwd, Center: Halfcheetah-pier, Right: Walker2D-randomparams

Left: Ant-fwd-bwd, Center: Ant-crippled-leg, Right: Humanoid-direc

**Figure 8:** Environments for our experiment

A.12    COMPLEMENTARY EXPERIMENTAL RESULTS

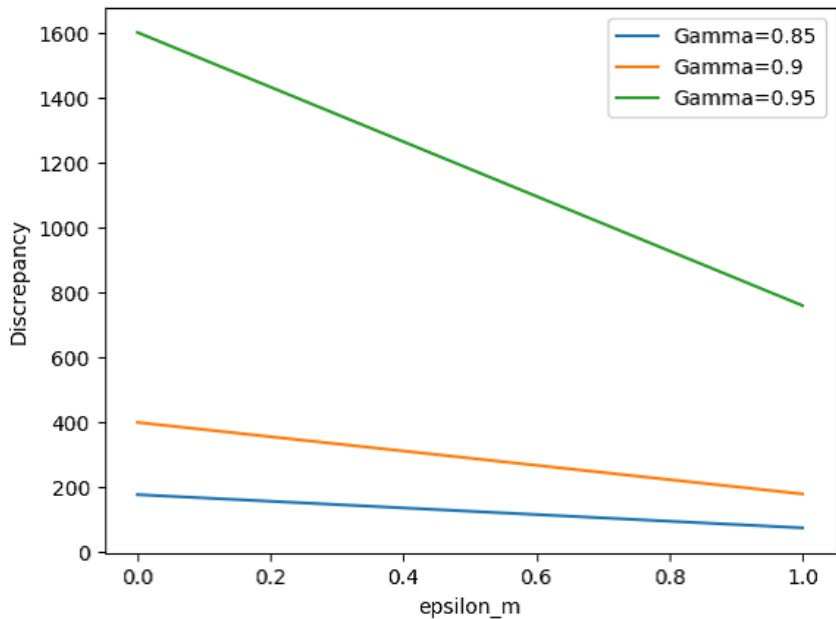

**Figure 9:** Discrepancy between $\mathbb{E}_{\pi_\phi, p}[R]$ and $\mathbb{E}_{\pi_\phi, p_\theta}[R]$ in Theorem 1. The vertical and horizontal axes represent the discrepancy value and $\epsilon_m \in [0, 1]$, respectively. We set the other variables as $\epsilon_\pi = 1 - \epsilon_m$ and $r_{\max} = 1$.

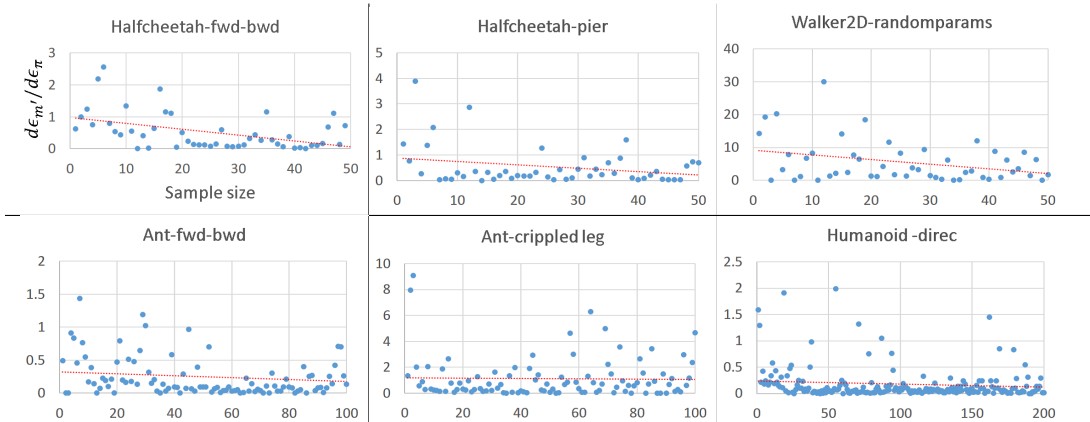

**Figure 10:** The local change in $\epsilon_{m'}$ with respect to $\epsilon_\pi$ versus training sample size. In each figure, the vertical axis represents the local change of the meta-model error ($\frac{d\epsilon_{m'}}{d\epsilon_\pi}$) and the horizontal axis represents the training sample size (x1000). The red-dotted line is the linear interpolation of the blue dots, which shows the trend of the local change decreasing as the training sample size grows.

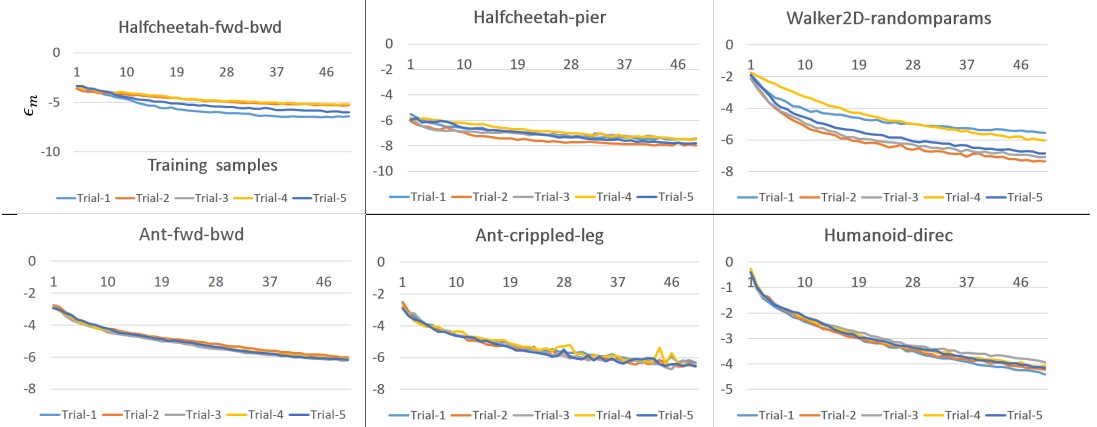

**Figure 11:** Transition of model errors on training. In each figure, the vertical axis represents empirical values of $\epsilon_m$ and the horizontal axis represents the number of training samples (x1000). We ran five trials with different random seeds. The result of the $x$-th trial is denoted by Trial-$x$. We used the negative of log-likelihood of the meta-model on validation samples as the approximation of $\epsilon_m$. The figures show that the model error tends to decrease as epochs elapse.

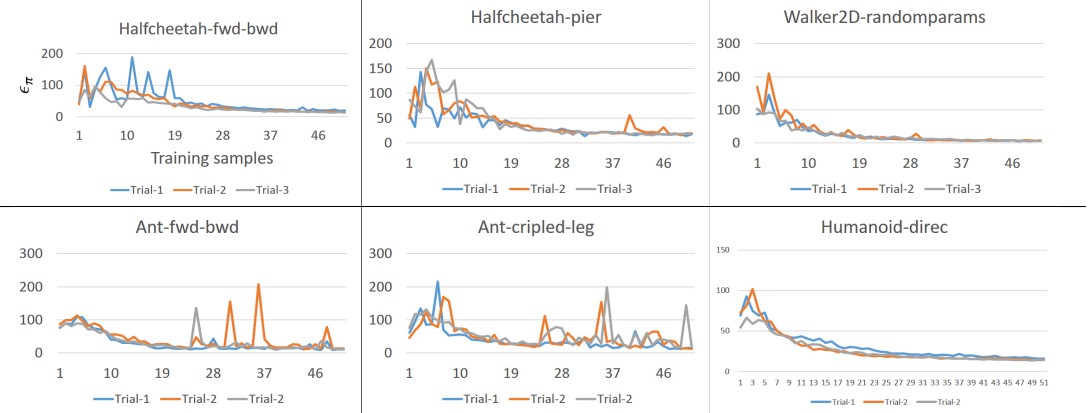

**Figure 12:** Transition of meta-policy divergence on training. In each figure, the vertical axis represents empirical values of $\epsilon_\pi$ and the horizontal axis represents the number of training samples (x1000). We ran three trials with different random seeds. The result of the $x$-th trial is denoted by Trial-$x$. For $\epsilon_\pi$, we used the empirical Kullback-Leibler divergence of $\pi_\theta$ and $\pi_\mathcal{D}$. Here, $\pi_\mathcal{D}$ has the same policy network architecture with $\pi_\theta$ and is learned by maximizing the likelihood of actions in $\mathcal{D}$ ($\mathcal{D}_{\text{env}}$ for Algorithm 2).

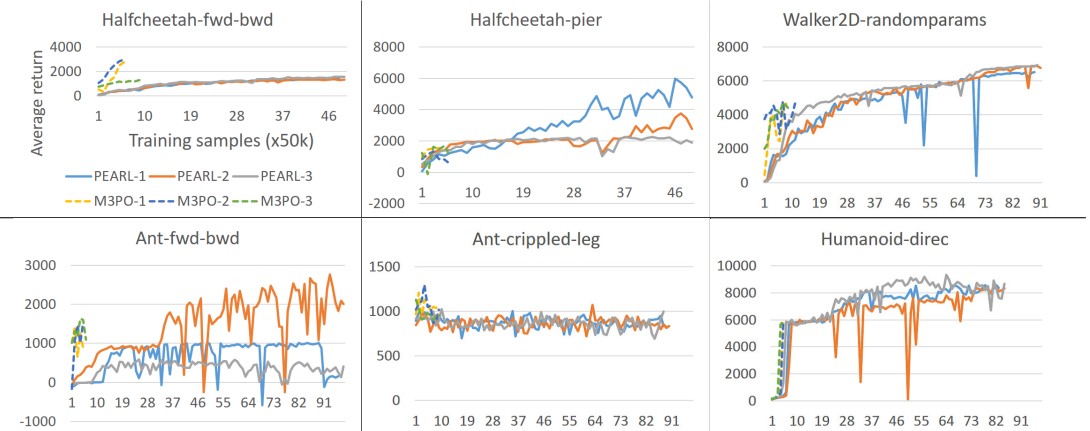

**Figure 13:** Learning curve of PEARL and M3PO in a long-term training. In each figure, the vertical axis represents expected returns and the horizontal axis represents the number of training samples (**x50000**). The meta-policy and meta-model were fixed and their expected returns were evaluated on 50 episodes at every 50,000 training samples. Each method was evaluated in three trials, and the result of the $x$-th trial is denoted by method-$x$. **Note that the scale of the horizontal axis is larger than that in Figure 1 by 50 times (i.e., 4 in this figure is equal to 200 in Figure 1).**

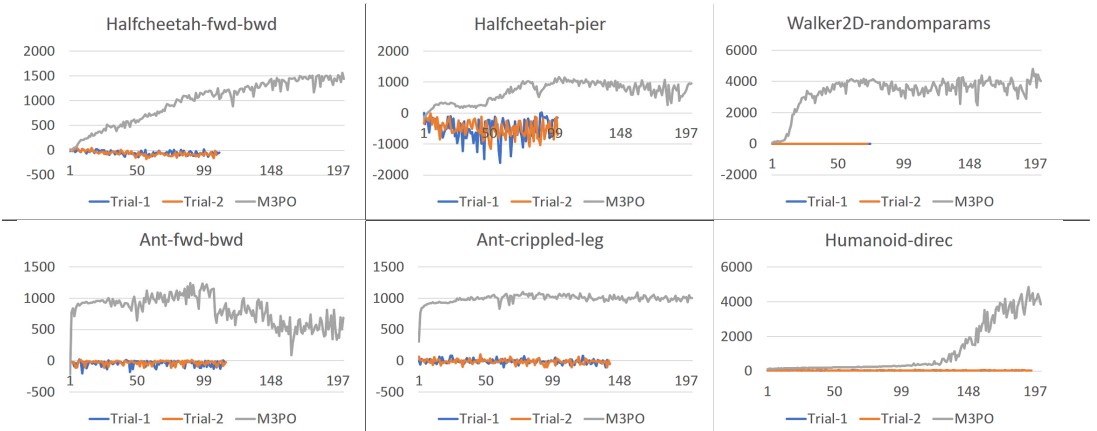

**Figure 14:** Comparison of GHP-MDP (Algorithm 1 in Perez et al. (2020)) and M3PO. The figures show learning curves of GHP-MDP and M3PO. In each figure, the vertical axis represents expected returns and the horizontal axis represents the number of training samples (x1000). GHP-MDP was evaluated in two trials, and each trial was run for three days in real-times. Due to the limitation of computational resources, we could not run this experiment as many days and trials as other experiments. The expected returns of GHP-MDP in each trial (denoted by "Trial-1" and "Trial-2") are plotted in the figure. The results of M3PO is referred to those in Figure 1. From the comparison result, we can see that M3PO achieves better sample efficiency than GHP-MDP.

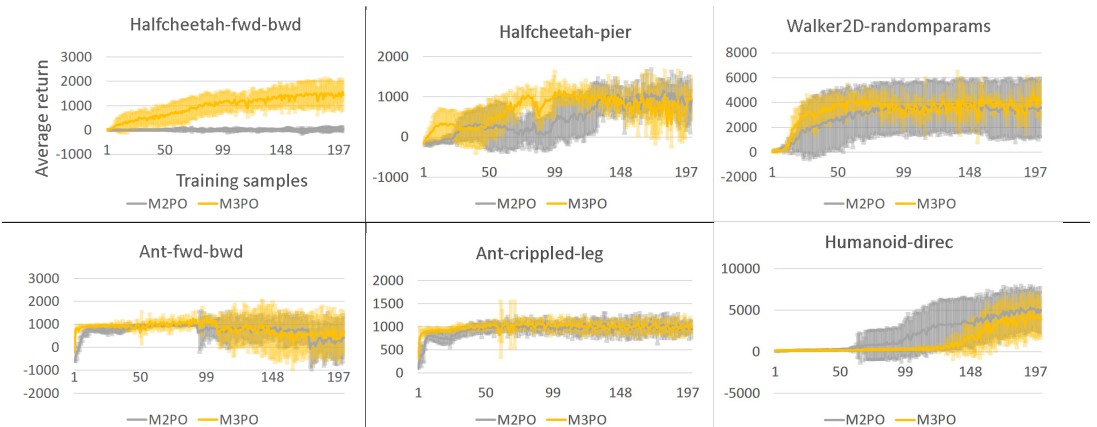

**Figure 15:** Learning curve of M3PO and M2PO. In each figure, the vertical axis represents expected returns and the horizontal axis represents the number of training samples (x1000). The meta-policy and meta-model (and a predictive model) were fixed and their expected returns were evaluated on 50 episodes at every 1000 training samples for the other methods. In each episode, the task was initialized and changed randomly. Each method was evaluated in at least five trials, and the expected return on the 50 episodes was further averaged over the trials. The averaged expected returns and their standard deviations are plotted in the figures.

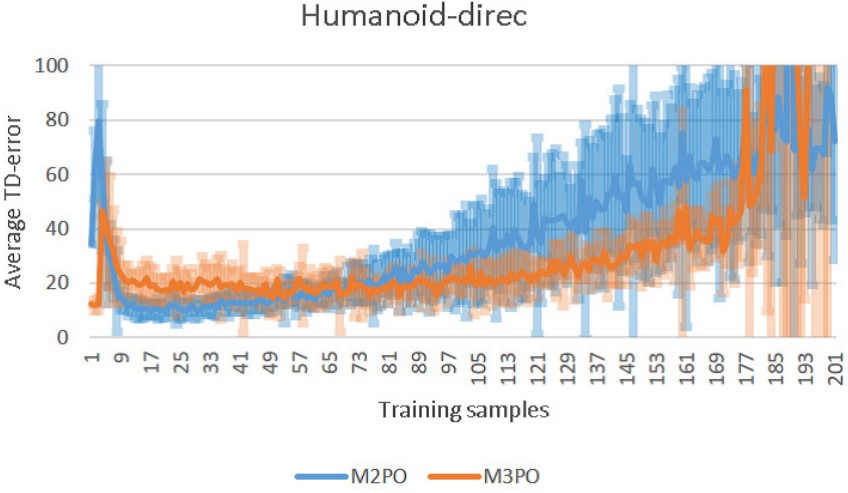

**Figure 16:** Transition of TD-errors (Q-function error) on training. In each figure, the vertical axis represents empirical values of $\epsilon_m$ and the horizontal axis represents the number of training samples (x1000). We ran ten trials with different random seeds and plotted the average of their results. The error bar means one standard deviation.

A.13 COMPLEMENTARY ANALYSIS

In addition to Q1 and Q2 in the main content, we also conducted a complementary analysis to answer the following question. **Q.3:** Does the use of a meta-model in M3PO contribute to the improvement of the meta-policy?

In an analysis in this section, we compare M3PO with the following method. **Model-based Meta-Policy Optimization (M2PO):** This method is a variant of M3PO, in which a non-adaptive predictive model is used instead of the meta-model. The predictive model architecture is the same as that in the MBPO algorithm (Janner et al., 2019) (i.e., the ensemble of Gaussian distributions based on four-layer feed-forward neural networks).

Regarding Q3, our experimental result indicates that the use of a meta-model contributed to the performance improvement in a number of the environments. In Figure 15 in the appendix, we can clearly see the improvement of M3PO against M2PO in Halfcheetah-fwd-bwd. In addition, in the Ant environments, although the M3PO's performance is seemingly the same as that of M2PO, the qualitative performance is quite different; the M3PO can produce a meta-policy for walking in the correct direction, while M2PO failed to do so (M2PO produces the meta-policy "always standing" with a very small amount of control signal). For Humanoid-direc, in contrast, M2PO tends to achieve better sample efficiency than M3PO. We hypothesize that the primary reason for this is that during the plateau at the early stage of training in Humanoid-direc, the predictive model used in M2PO generates fictitious trajectories that make meta-policy optimization more stable. To verify this hypothesis, we compare TD-errors (Q-function errors) during training, which is an indicator of the stability of meta-policy optimization, in M3PO and M2PO. The evaluation result (Figure 16 in the appendix) shows that during the performance plateau (10–60 epoch), the TD-error in M2PO was actually lower than that in M3PO; this result supports our hypothesis. In this paper, we did not focus on the study of meta-model usage to generate the trajectories that make meta-policy optimization stable, but this experimental result indicates that such a study is important for further improving M3PO.

A.14 HYPERPARAMETER SETTING

**Table 1:** Hyperparameter settings for M3PO results shown in Figure 1. $x \rightarrow y$ over epohs $a \rightarrow b$ denotes a thresholded linear function, i.e., at epoch $e$, $f(e) = \min(\max(x + \frac{e-a}{b-a} \cdot (y - x), x), y)$.

| | | Halfcheetah-fwd-bwd | Halfcheetah-pier | Ant-fwd-bwd | Ant-crippled-leg | Walker2D-randomparams | Humanoid-direc |
|---|---|---|---|---|---|---|---|
| $N$ | epoch | \multicolumn{6}{c}{200} | | | | | |
| $E$ | environment step per epoch | 1000 | | | | | |
| $M$ | meta-model rollouts per environment step | 1e3 | | | 5e2 | 1e3 | 5e2 |
| $B$ | ensemble size | 3 | | | | | |
| $G$ | meta-policy update per environment step | 40 | | 20 | | | |
| $k$ | meta-model horizon | 1 | 1 | 1 → 25 over epoch 20 → 100 | | 1 | 1 |

A.15 NOTATIONS

**Table 2:** List of mathematical notation in the main body. For simplicity, we do not include notation introduced in the appendix.

| Notation | Meaning |
|---|---|
| $t$ | Time step |
| $o_t$ | Observation |
| $\mathcal{O}$ | Set of observations |
| $s_t,\ s',\ s'_t$ | Hidden state |
| $\mathcal{S},\ \mathcal{S}'$ | Set of hidden states |
| $a,\ a_t$ | Action |
| $\mathcal{A}$ | Set of actions |
| $p_{\text{ob}},\ p(o_t|s_t, a_{t-1})$ | Observation probability function |
| $p_{\text{st}},\ p(s_t|s_{t-1}, a_{t-1})$ | State transition probability function |
| $r_t,\ r(s_t, a_t)$ | Reward function |
| $\gamma$ | Discount rate |
| $\pi,\ \pi_\phi,\ p(a_{t+1}|h_t)$ | (Meta-)policy |
| $h,\ h_t,\ h'$ | History (past trajectories) |
| $\mathcal{H}$ | Set of history |
| $\phi$ | Learnable parameters of meta-policy |
| $\theta$ | Learnable parameters of meta-model |
| $p(s_t|h_t)$ | Belief state |
| $R$ | Discounted return |
| $\mathbb{E}_{a\sim\pi_\phi, r, h\sim p_\theta}[R],\ \mathbb{E}_{\pi_\phi, p_\theta}[R]$ | Meta-model return under full meta-model rollout |
| $C$ | Discrepancy between the meta-model return and true return |
| $\epsilon_m$ | Bounds of generalization error |
| $\epsilon_\pi$ | Bounds of distribution shift |
| $\mathcal{D},\ \mathcal{D}_{\text{env}}$ | Dataset (trajectories collected from the real environment) |
| $D_{TV}$ | Total variation distance |
| $r_{max}$ | Bound of the reward expected on the basis of a belief |
| $\mathcal{D}_{\text{model}}$ | Set of samples collected through branched rollouts |
| $k$ | Meta-model horizon (meta-model rollout length) |
| $\pi_\mathcal{D}$ | Data collection policy |
| $\mathbb{E}_{\pi_\phi, p}[R]$ | True return (the return of the meta-policy in the real environment) |
| $\epsilon_{m'}$ | Bound of meta-model error on a current policy |
| $\mathbb{E}_{(a,h)\sim\mathcal{D}_{\text{model}}}[R]$ | Meta-model return in branched meta-rollouts |
| $z$ | Latent context |
| $p_\phi(z|a_0, o_0, ..., a_t, o_t)$ | History enoder |
| $D_{KL}$ | Kullback-Leibler divergence |
| $J_{\mathcal{D}_{\text{model}}}$ | Optimization objective for meta-policy optimization in M3PO |
| $Q_{\pi_\phi}$ | Action value function |
| $V_{\pi_\phi}$ | State value function |
| $B$ | Ensemble size |
| $i$ | Index |
| $\mathcal{N}$ | Gaussian distribution |
| $\mu_\theta^i$ | Mean of Gaussian distribution |
| $\sigma_\theta^i$ | Standard deviation of Gaussian distribution |
| $\tau,\ \tau_t$ | Task |
| $\mathcal{T}$ | Task set |

**Table 3:** $\mathbb{E}_k[k = 1]$ settings for results shown in Figure 17. $a \to b$ denotes a thresholded linear function, i.e., at epoch $e$, $f(e) = \min(\max(1 - \frac{e-a}{b-a}, 1), 0)$.

| | | Halfcheetah-pier | Walker2D-randomparams | Humanoid-direc |
|---|---|---|---|---|
| $\mathbb{E}_k[k = 1]$ | expected meta-model horizon | $80 \to 130$ | $50 \to 100$ | $150 \to 250$ |

## A.16 COMPLEMENTARY ANALYSIS 2

In Figures 1 and 13, we can see that, in a number of the environments (Halcheetah-pier, Walker2D-randomparams and Humanoid-direc), the long-term performance of M3PO is worse than that of PEARL. This indicates that a gradual transition from M3PO to PEARL (or other model-free approaches) needs to be considered to further improve overall performance. In this section, we propose to introduce such a gradual transition approach to M3PO and evaluate it on the environments where the long-term performance of M3PO worse than that of PEARL.

For the gradual transition, we introduce the notion of the "expected" model-rollout length $\mathbb{E}_k[k]$ for $k \in \{0, 1\}$ to M3PO. In this notion, $\mathbb{E}_k[k = 1]$ means the probability of one-step meta-model rollout. Namely, with the probability of $\mathbb{E}_k[k = 1]$, the fictitious trajectory generated by one-step meta-model rollout is used for the policy update (in line 11 in Algorithm 2), and, with the probability of $1 - \mathbb{E}_k[k = 1]$, the real trajectory in $\mathcal{D}_{\text{env}}$ are used for the policy update. In our implementation, we replace $\mathcal{D}_{\text{model}}$ in line 11 in Algorithm 2 with the mixed dataset $\mathcal{D}_{\text{mix}}$. Here, $\mathcal{D}_{\text{mix}}$ is defined as

$$\mathcal{D}_{\text{mix}} = \mathbb{E}_k[k = 1] \cdot \mathcal{D}_{\text{model}} + (1 - \mathbb{E}_k[k = 1]) \cdot \mathcal{D}_{\text{env}}. \tag{38}$$

We linearly reduce the value of $\mathbb{E}_k[k = 1]$ from 1 to 0 depending on training epoch. With this value reduction, the M3PO is gradually transitioned to PEARL (i.e., the model-free approach).

We evaluate M3PO with the gradual transition (M3PO-h) in three environments (Halfcheetah-pier, Walker2D-randomparams and Humanoid-direc) where the long-term performance of M3PO is worse than that of PEARL in Figures 1 and 13. The hyperparameter setting (except for setting schedule for the value of $\mathbb{E}_k[k = 1]$) for the experiment is the same as that for Figures 1 and 13 (i.e., the one shown in Table 1). Regarding to setting schedule for the value of $\mathbb{E}_k[k = 1]$, we reduce the value of $\mathbb{E}_k[k = 1]$ in accordance with Table 3.

Evaluation results are shown in Figure 17. Due to the limitation of our computational resource, we could not continue the training of M3PO-h until convergence, and the resulting training epoch is much shorter than those of PEARL. Nevertheless, we can see that some trials of M3PO-h achieve same or better scores with the long-term performances of PEARL in all environments (e.g., M3PO-h-1 and M3PO-h-2 achieve the same or better performance than the best scores of PEARL-2 and PEARL-3 in Halfcheetah-pier).

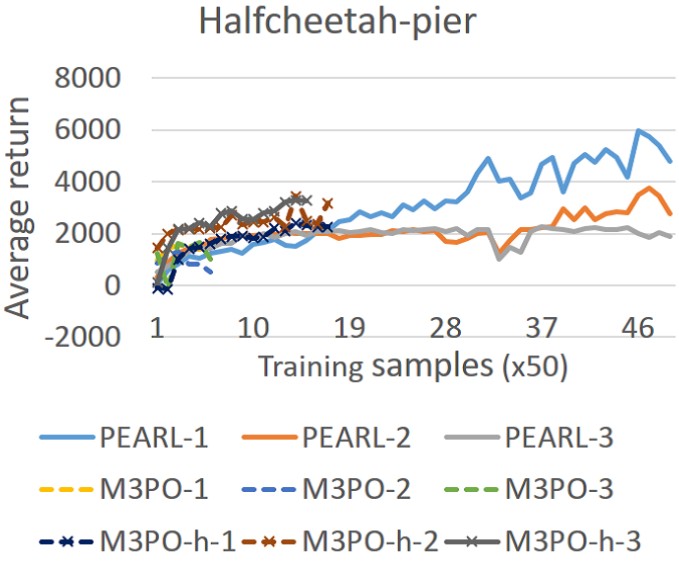

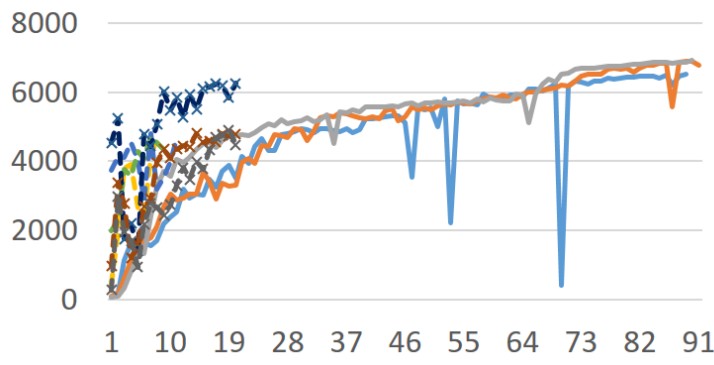

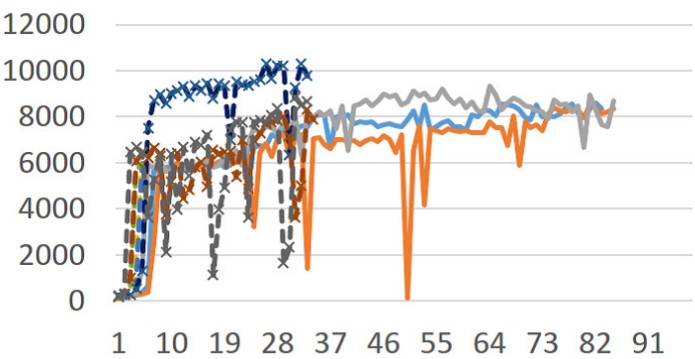

**Figure 17:** Learning curve of PEARL, M3PO and M3PO-h in a long-term training. In each figure, the vertical axis represents expected returns and the horizontal axis represents the number of training samples (**x50000**). The meta-policy and meta-model were fixed and their expected returns were evaluated on 50 episodes at every 50,000 training samples. Each method was evaluated in three trials, and the result of the $x$-th trial is denoted by method-$x$. **Note that the scale of the horizontal axis is larger than that in Figure 1 by 50 times (i.e., 4 in this figure is equal to 200 in Figure 1).**

## A.17    HIGH-LEVEL SUMMARY

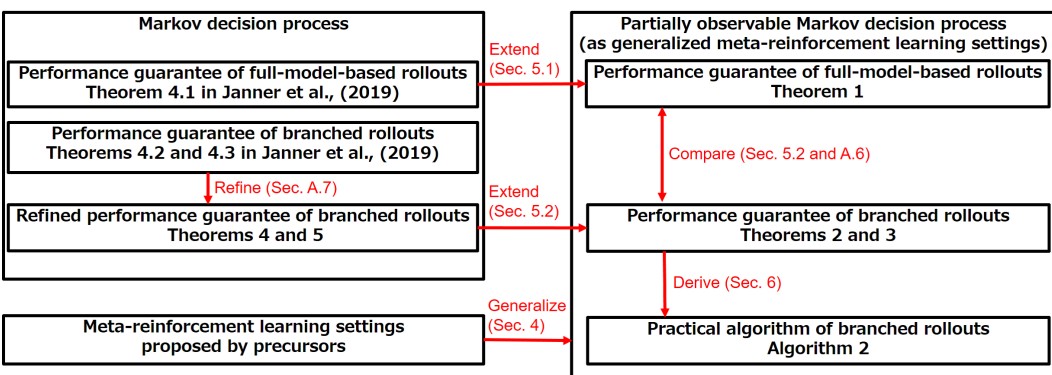

**Figure 18:** High-level summary of what we do in this paper based on previous works. Our core contribution is proposing a model-based meta-reinforcement learning (RL) algorithm (i.e., Algorithm 2) with performance guarantee. To achieve this, we first generalize various existing model-based meta-RL settings as solving a partially observable Markov decision process (POMDP) (Section 4). We then conduct theoretical analysis by extending the theorems proposed in Janner et al. (2019). We extend their theorems (Theorems 4.1, 4.2 and 4.3) into our POMDP setting (Sections 5.1 and 5.2). As a part of our extension, we refine Theorem 4.2 and 4.3 in order to strictly guarantee the performance of branched rollouts (Section A.7). We also compare these theorems in order to motivate using branched rollouts for model-based meta-RL (Sections 5.2 and A.6). We finally propose a practical algorithm based on the result of our theoretical anyways (Section 6).

