# OpenReview forum: "Meta-Model-Based Meta-Policy Optimization"
_ICLR.cc/2021/Conference — Reject_

### Official Review · AnonReviewer4 · 2020-10-27
**Fair paper - Straightforward extension of Janner et al. (2019) to POMDPs**

**Rating:** 5
**Confidence:** 3

**Review:**

=== Summary ===

The paper concerns model-based meta-RL. It exploits the fact that meta-RL can be formulated as POMDP in which the task indicator is part of the (unobserved) hidden state. Thus, the paper effectively analyzes and proposes model-based algorithms for POMDPs. The paper bounds the gap between the expected reward of a policy in the actual POMDP and the estimated model and then theoretically shows that this gap can be reduced when using dyna-style / branched rollouts instead of full rollouts under the learned model. Motivated by this finding, the paper proposes a Dyna-like algorithm for POMDPs. In the experimental evaluation, the paper compares its proposed method, M3PO, to two recent meta-RL approaches in a range of meta-RL environments for continuous control.

=== Merits of the paper ===

Overall, the paper contributes a new algorithm for model-based meta-RL which is neatly motivated by the paper’s theoretical analysis. Both the theoretical analysis and the proposed algorithm are sound. The experimental evaluations demonstrate that the algorithm performs similar or better than previous model-based meta-RL algorithms and thus could be a relevant contribution to the field of meta-RL.

=== Strenghts ===
- The experiments include relevant meta-learning environments and algorithms to compare to.
- The proposed algorithm is sound and motivated by theory.
- I have done a very simple simulation study, being able to confirm that the gap in Theorem 2 is indeed (much) smaller than the one Theorem 1.

=== Weaknesses / Concerns ===
- Both Theorem 1 and 2 seem to be a straightforward combination of the results in [1] and the fact that POMDPs can be cast as MDPs with history-states. Thus the theoretical contribution/novelty is quite limited.
- As expected, from my simulation study it seems that bounds are vacuous and their usefulness beyond motivating branched rollouts is questionable.
- The only difference to original dyna-like approaches is the fact that a recurrent model with internal/hidden state is used – thus the algorithmic contribution is small as well.
- Overall, the clarity of the paper could probably be improved a lot – Many paragraphs and sentences are hard read and lack vital explanations. For examples, see below.

=== Overall Assessment ===

In my opinion, the paper is a borderline case. Currently, I see the paper slightly below the acceptance threshold. Both the theoretical and the algorithmic contribution of the paper are small. Overall, the paper is a straightforward extension of [1] to POMDPs with meta-RL experiments. Yet, due to a lack of clarity, it is hard to read. Nonetheless, the proposed algorithm seems to be practically relevant for meta-RL - thus I am happy to hear other opinions and open to being convinced to increase my score.

=== Questions & tips for improvement ===

Appendix A.1 briefly discusses the differences to [1], and claims to derive similar Theorems under more appropriate assumptions / premises in Appendix A.7. How do the theorems in A.7. differ from the ones in Section 5? If the theorems build on more appropriate assumptions, why not use them in the main part of the paper?

Adding a simple simulation study that shows how much smaller the gap/discrepancy is, when branched instead of full rollouts are used, would require little effort and strengthen the claim that branched rollouts are preferable.

I found section 5.1. to be very confusing – It is stated that the gap can be “expressed as the function of two error quantities of the meta-model: generalization error due to sampling and distribution shift due to the updated meta-policy”. Indeed Def. 2 is probably the generalization error due to sampling. However, Def. 1 is the expected TV distance of the estimated model from the true model. Then, how should this be the “distribution shift due to the updated meta-policy”?

=== Minor comments ===

- Section 4, 3rd paragraph: It should probably be "$r_{t}$ and $o_{t+1}$ are assumed to be conditionally independent
- Section 5.1.: It would be good to have a proper definition if $\pi_{\mathcal{D}}$, i.e. the data-collection policy
- Theorem 1: The definition of $\epsilon_m$ is duplicate whereas the $\epsilon_\pi$ is not included
- Section 7, 4th paragraph: Should be “In Humanoid-direct, the performance”


[1] Michael Janner, Justin Fu, Marvin Zhang, and Sergey Levine. When to trust your model: Model-based policy optimization. NeurIPS 2019

---

> ### Author Response · Authors · 2020-11-19
> **Our response to Reviewer 4's review**
>
> Thank you very much for your comments.
> We posted our response to common concerns on our work, at the top of this board.
>
> Here is our response to your comments:
>
> > Both Theorem 1 and 2 seem to be a straightforward combination of the results in [1] and the fact that POMDPs can be cast as MDPs with history-states. Thus the theoretical contribution/novelty is quite limited.
>
> Although it is relatively straightforward to derive Theorem 1, we would like to argue that Theorem 2 is not a trivial and straightforward extension of [1].
> This is because the proof of our theorems requires a refined version of the theorems in [1] that bounds the performance of branched rollouts more properly (that's what we derived in A.7).
> Therefore, our contribution is providing not only the bound of the branched rollouts performance for POMDPs, but also the proper bound of the branched rollout performance.
> (see also our reply to Reviewer 5)
>
>
> > Appendix A.1 briefly discusses the differences to [1], and claims to derive similar Theorems under more appropriate assumptions/premises in Appendix A.7. How do the theorems in A.7. differ from the ones in Section 5? If the theorems build on more appropriate assumptions, why not use them in the main part of the paper?
>
> Theorem 4 in A.7 is a theorem for MDPs, and Theorem 2 in Section 5 is its extension for POMDPs.
> We did not include theorems in A.7 in the main part due to space limitation.
> We will include them into the main part to improve the clarity.
>
> > Adding a simple simulation study that shows how much smaller the gap/discrepancy is, when branched instead of full rollouts are used, would require little effort and strengthen the claim that branched rollouts are preferable.
>
> We had conducted a simulated study, in which the trend of discrepancy values in the full-rollout case (Theorem 1) and that in branched-rollout case (Theorems 2 and 3) are evaluated.
> The trend of the performance discrepancy values in Theorem 1 is shown in Figure 9 and that in Theorems 2 and 3 is shown in Figure 5.
> Given appropriate values for $k$ for Theorem 2 and 3, the results in those figures indicate that the discrepancy in the branched-rollout case is smaller than that in the full-rollout case.
>
> In addition, from the viewpoint of empirical performance discrepancy (i.e., the discrepancy between a training performance in the meta-model and an actual performance in the true environment), we will evaluate M3PO against the variant of M3PO that uses full rollouts.
> We expect that the empirical performance discrepancy of M3PO is smaller than that of the variant.
>
>
>
> > I found section 5.1. to be very confusing – It is stated that the gap can be “expressed as the function of two error quantities of the meta-model: generalization error due to sampling and distribution shift due to the updated meta-policy”. Indeed Def. 2 is probably the generalization error due to sampling. However, Def. 1 is the expected TV distance of the estimated model from the true model. Then, how should this be the “distribution shift due to the updated meta-policy”?
>
> Def. 1 is the expected TV distance between the true environment and the model. Therefore, it measures the error occurs by sampling from the model.
> Def. 2 is the TV distance between the data collection policy and the learning policy. Therefor it measures error due to the distributional shift.
>
> We will revise this part as follows:
> "The discrepancy between these returns, $C$, can be expressed as the function of two error quantities of the meta-model: generalization error due to sampling and distribution shift due to the updated meta-policy."
> ->
> "The discrepancy between these returns, $C$, can be expressed as the function of two error quantities: the generalization error of the meta-policy and the distribution shift due to the updated meta-policy."

---

### Official Review · AnonReviewer3 · 2020-10-27
**Theory improvement and small extension of existing work**

**Rating:** 5
**Confidence:** 3

**Review:**




## General Summary

The contributions of this paper are two-fold. For one, the authors propose a meta model-based RL algorithm working with a meta-policy and a meta-model. In addition, the authors provide a lower bound of the meta-policy performance.
The algorithm is compared against PEARL and L2A, there is a further comparison against M2PO in the appendix.

## Writeup

I find the writeup quite dense and difficult to follow. For one, the idea behind meta-learning is generalization across different tasks/environments. While this was introduced well in sections 3 and 4, the connection is missing in the experiment section and algorithm section. The notation can be better introduced.

## Pros

The theoretical analysis is the greatest asset of this paper, it is useful to have a bound on the actual performance of the policy that depends on the length of the model rollout, model error and policy distribution shift. Also, it is a valid extension of MBPO by Jenner et al. (2019).

## Cons

The policy distribution shift and model error based bounds have been addressed in the model-based RL literature. What separates this paper is the extension to POMDPs and meta-learning formalism. The writeup is dense, although the idea behind is simple (branched model rollouts in POMDPs).

## Comments

Maybe it's because I don't work on meta-learning, but I think that you are really missing a clear description of the algorithm. Where are the task switches happening, how does the initial task distribution come into play? This should be clear from algorithm 2 in my opinion.

In Algorithm 2. you could introduce a loop over tasks maybe? I realize that in the environments that you used, task switches happen  during the episode, but I wouldn't rely on that in the algorithm description because it's a bit confusing, looking at the the algorithm itself, I don't see a meta-learning algorithm but rather a model-based RL algorithm for POMDPs.

The result that performance degrades with increasing model rollout horizon is not surprising and is the consequence of error compounding. This is a well-known result coming from standard model-based RL and MPC literature.

Branched rollouts (although meta-branched as the authors name them) are also also not a novel addition, as the authors note in the paper. The improvement was introduced in earlier literature (Dyna, MBPO).

Couple of questions for the experiment section:

* Are the results for L2A correct? They seem to be too bad to be true based on the results reported in the original paper.
* In Fig. 1, by number of training samples it is meant number of time steps in the environment?
* The variance of the performance is quite high for the ant fwd-bwd, much higher that the other methods. Any explanation for that?
* In Fig. 1, the legend should appear for the whole figure and not just for the upper-left graph.
* The performance of PEARL with long-run is better in 3/6 experiments and seems comparable in the others, does this mean that M3PO doesn't use new data efficiently? Also, I would be interested in knowing how the variance looks like for those longer runs.
* In Fig. 2, it is not really surprising that the performance degrades so much with the rollout length. What is surprising to me is that oftentimes the performance is best for k=1, indicating that there is not much use of the model except for 1 time step, this is in contrast to Janner et al. (2019) in the non-meta setting.
* I would be interested in seeing how the method generalizes across  OOD unseen tasks (not seen on train), such that the training and test distributions are different and then plot training vs test performance.

It's good that the authors make a clear comparison between this work and Jenner et al. (2019) because there is a lot of overlap and as I understand, code was reused. Nevertheless, I am concerned about the amount of novelty,
this is an extension to POMDPs with a correction to the theory.

Is the meta-learning formalism even needed for this kind of algorithm and evaluation?

Text-intricacies:
* check that the references are correct, p. 10 you have a reference of the form:  MBPO. [link]
* sec. 3 par. 2 polity->policy

---

> ### Author Response · Authors · 2020-11-19
> **Our response to Reviewer 3's review (1/2)**
>
> Thank you very much for your comments.
> We posted our response to common concerns on our work, at the top of this board.
>
> Here is our response to your comments:
>
>
> > Maybe it's because I don't work on meta-learning, but I think that you are really missing a clear description of the algorithm. Where are the task switches happening, how does the initial task distribution come into play? This should be clear from algorithm 2 in my opinion.
>
> The task switches in accordance with the transition function of the task when the trajectory is sampled from the environment in line 5 in Algorithm 2.
> Also, in line 5, the task is sampled from the initial task distribution at the beginning of an episode.
> Note that since the task information is assumed to be unobservable from the agent in our setting (see Section 4), the sampled task is used only in the environment and is not fed to the agent (algorithm).
>
> We will revise the paper to improve the clarity on this part.
>
>
> > In Algorithm 2. you could introduce a loop over tasks maybe? I realize that in the environments that you used, task switches happen during the episode, but I wouldn't rely on that in the algorithm description because it's a bit confusing, looking at the the algorithm itself, I don't see a meta-learning algorithm but rather a model-based RL algorithm for POMDPs.
>
> We answer your question, assuming that you are referring to the intra-task loop such as lines 3--9 in Algorithm 1 of the PEARL [3] paper as "a loop over tasks."
> We have not introduced such a task loop into Algorithm 2, since task information is assumed to be unobservable from the agent.
> Namely, in our algorithm description (M3PO), the trajectories sampled from the environments are not stored into a replay buffer for each task as in the original PEARL algorithm, but stored together into a single replay buffer.
>
>
> > Are the results for L2A correct? They seem to be too bad to be true based on the results reported in the original paper.
>
> We used the public source code of L2A published by the authors and tuned its hyper-parameters to our experimental setting.
>
> The results of L2A are different probably because we use a different (and more difficult) experimental setup.
> For example, in our "Ant-crippled-leg" environment, we introduce early termination (terminating an episode when the ant falls down) and allows the part of ant's crippled leg to change.
>
>
>
>
> > In Fig. 1, by number of training samples it is meant number of time steps in the environment?
>
> Yes.
>
> > The variance of the performance is quite high for the ant fwd-bwd, much higher that the other methods. Any explanation for that?
>
> Here is our hypothesis based on the observation of the meta-policies' behaviour during the learning:
> It is quite difficult to make the ant correctly move forward and backward in accordance with the given task, and it makes the meta-policy unstable.
> In Ant-fwd-bwd, the learning agent tends to at first learn to move forward correctly when the task is "forward" and stop when the task is "backward" (or vice versa). At this first step, learning is stable (so the variance of performance is relatively small until around the first 100 episodes). After that, the agent starts to learn to correctly move forward and backward in accordance with the task, but it is difficult to learn and makes the learning unstable.
> Note that the performance of other methods is more stable because their policy could not learn this task at all.
>
> > In Fig. 1, the legend should appear for the whole figure and not just for the upper-left graph.
>
> Thank you for the suggestion. We will modify the paper as you suggested.
>
> > The performance of PEARL with long-run is better in 3/6 experiments and seems comparable in the others, does this mean that M3PO doesn't use new data efficiently?
>
> This is primarily due to unstable learning incurred by meta-model rather than inefficient data leverage.
> Since M3PO uses the fictitious data generated by the meta-model rollout to train the meta-policy (and value function), the performance improvement seems to be more sensitive to meta-model ambiguity and distributional shifts due to meta-model updates as the training progresses.
> To reduce the reliance on meta-model, we modified M3PO to use a mixture of real and fictitious data.
> The evaluation results of the modified M3PO are shown in the revised paper A.16.
> The results indicated that the long-term performance of M3PO with the mixed data is comparable to PEARL.
>
> [3] Kate Rakelly, Aurick Zhou, Chelsea Finn, Sergey Levine, and Deirdre Quillen.  Efficient off-policy meta-reinforcement learning via probabilistic context variables. ICML, 2019.

---

> > ### Author Response · Authors · 2020-11-19
> > **Our response to Reviewer 3's review (2/2)**
> >
> > > Also, I would be interested in knowing how the variance looks like for those longer runs.
> >
> > As for PEARL-long, the variances in Halfcheetah-pier and Ant-fwd-bwd are large compared to those in rest of the environments. In Halfcheetah-pier and Ant-fwd-bwd, the learning progresses in each trial are quite different depending on initial random-seed.
> > As for M3PO-long, the variances are smaller than those in PEARL-long in most of the environments because the best scores in each trial are nearly the same.
> > Figure 13 provides the results in  each trial for both PEARL-long and M3PO-long.
> >
> > > I would be interested in seeing how the method generalizes across OOD unseen tasks (not seen on train), such that the training and test distributions are different and then plot training vs test performance.
> >
> > We will conduct an additional experiment for this. We hypothesize that M3PO is able to generalize across the OOD tasks proposed in the PEARL paper since the PEARL architecture is used in our algorithmic implementation.
> >
> >
> > > For one, the idea behind meta-learning is generalization across different tasks/environments. While this was introduced well in sections 3 and 4, the connection is missing in the experiment section and algorithm section.
> > > Is the meta-learning formalism even needed for this kind of algorithm and evaluation?
> >
> > The algorithms and evaluation do not require the meta-learning formalism. However, our research focuses on meta-RL, and we believe that the meta-learning formulation in Section 4 is necessary in order to show that the theorems we derived in Section 5 can be applied to it.

---

### Official Review · AnonReviewer2 · 2020-10-30
**Limited novelty compared to the existing works**

**Rating:** 5
**Confidence:** 3

**Review:**

### Paper summary

This paper focus on model-based RL on a POMDP setting (they call it "meta RL"), where the policy and model need to infer the current hidden state according to history. It provides a theoretical relation between true environment returns and the returns from learned models in a POMDP setting. And it also provides a practical algorithm called M3PO and shows this algorithm is more sample efficient than some meta-RL baselines in some continuous control tasks.

### Pros

- Extend an existing work ( Janner et al. (2019) ) to a POMDP setting
- Refine some theorems in Janner et al. (2019), by taking more important premises into consideration

### Cons

- The theory part of this paper is quite similar to Janner et al. (2019). It seems they just apply similar theorems to the POMDP setting. For example, theorem 1 in this paper is an immediate result when combining Theorem 4.1 in Janner et al. (2019) and Lemma 1 in Silver & Veness (2010). And the idea of using branched rollout to control model errors is not first proposed by this paper either.
- The proposed algorithm (M3PO) is also very similar to algorithm 2 in Janner et al. (2019), and they just use this algorithm in POMDPs. From an implementation perspective, there are only two differences: 1. using GRU as network architectures, 2. use PEARL in policy optimization. So I feel the novelty in the proposed algorithm is limited.
- In the experiments, it seems M3PO (their algorithm) only has significant advantages over PEARL on Halfcheetah-fwd-bwd and Ant-fwd-bwd. In the other environments, the performance gap seems not very clear. Also, in Fig 1, they only show the horizon of 0.2 M steps, which is a relatively short training time for these difficult tasks. And if we compare M3PO-long and PEARL-long, it seems M3PO-long is overall worse than PEARL-long.

### Questions

- Why MBPO ( Janner et al. (2019) ) does not support POMDP? Just because it does not use RNN?

### Minor points:

- The figures are pretty hard to read, especially fig 2.

### References

- Michael Janner, Justin Fu, Marvin Zhang, and Sergey Levine. When to trust your model: Model-based policy optimization. In Proc. NeurIPS, 2019.
- David Silver and Joel Veness. Monte-Carlo planning in large POMDPs. In Proc. NIPS, pp. 2164– 2172, 2010.

---

> ### Author Response · Authors · 2020-11-19
> **Our response to Reviewer 2's review**
>
> Thank you very much for your comments.
> We posted our response to common concerns on our work, at the top of this board.
>
> Here is our response to your comments:
>
>
> > The theory part of this paper is quite similar to Janner et al. (2019). It seems they just apply similar theorems to the POMDP setting. For example, theorem 1 in this paper is an immediate result when combining Theorem 4.1 in Janner et al. (2019) and Lemma 1 in Silver & Veness (2010). And the idea of using branched rollout to control model errors is not first proposed by this paper either.
>
> As for the derivation part of Theorem 1, as you pointed out, we simply applied the known theorem and lemma.
> However, as for the derivation part of Theorems 2 and 3, we use the refined version of the theorems in Janner et al. (2019) that bound the performance of branched rollouts more properly (Theorems 4 and 5 in A.7).
> In this respect, we do more than just apply the known theorem and lemma.
> (See also our reply to Reviewer 5)
>
>
> > The proposed algorithm (M3PO) is also very similar to algorithm 2 in Janner et al. (2019), and they just use this algorithm in POMDPs. From an implementation perspective, there are only two differences: 1. using GRU as network architectures, 2. use PEARL in policy optimization. So I feel the novelty in the proposed algorithm is limited.
> > Why MBPO ( Janner et al. (2019) ) does not support POMDP? Just because it does not use RNN?
>
> MBPO does not support POMDPs because of more algorithmic-level differences:
> 1. MBPO performs the model rollout based on (MDP) states, not the history of (POMDP) observations. More specifically, MBPO randomly samples a (MDP) state contained in D_{\text{env}} during the model rollout (line 8 in their Algorithm 2), and, in this, the sequential information is not taken into account.
> 2. The type of input to the policy and transition model is not the history. MBPO treats the (MDP) state as the type of their input, which is different from the history of observations (i.e., h in our paper).
> The use of RNNs (for meta-policy and meta-model) is an instance of approaches to address the second point. (i.e., there are other approaches. For example, a gradient-based MAML architecture can be used for the meta-policy and meta-model instead.)
> Also, to address the first point, modification of D_{\text{env}} is necessary.
>
> As for the implementation, note that, in the last paragraph of Section 6, we explain only the main extensions from the MBPO implementation described, and we have made other extensions.
> For example, we introduce gradient clipping to deal with training instability derived from the nature of the sequential model, and dynamic automated handling to deal with the breakdown of meta-model training.
>
> > In the experiments, it seems M3PO (their algorithm) only has significant advantages over PEARL on Halfcheetah-fwd-bwd and Ant-fwd-bwd. In the other environments, the performance gap seems not very clear.
>
> In terms of sample efficiency, M3PO has significant advantages over PEARL on Halfcheetah-fwd-bwd, Walker2D-randomparams, Ant-fwd-bwd, Ant-crippled-leg and Humanoid-direc, because, in Figure 1, the learning curves of M3PO are plotted at upper left parts than those of PEARL.
> (i.e., In these environments, M3PO achieve a certain level of performance significantly faster than PEARL.)
>
>
> > And if we compare M3PO-long and PEARL-long, it seems M3PO-long is overall worse than PEARL-long.
>
> Since M3PO only uses fictitious data generated by the meta-model rollout, it tends to be defeated by the model-free RL in the long-term training.
> As an additional experiment, we have evaluated a modified version of M3PO that uses a mixture of the fictitious and real data for meta-policy update.
> The results of this experiment are given in Appendix A.16 of the revised version. The results show that the long term performance of M3PO with a mixture data is comparable to that of PEARL.

---

### Official Review · AnonReviewer5 · 2020-11-04
**Seems like the TL;DR is a modification of (Janner, 2019)'s method to use an RNN model for meta-learning.**

**Rating:** 6
**Confidence:** 3

**Review:**

### **Summary and Contributions of Paper**
This paper modifies the work "When to Trust Your Model: Model-Based Policy Optimization" (Janner, 2019) to handle the meta-learning case. The definitions, paper structure, and algorithm flows are all very similar to (Janner, 2019). The main difference is that the model architecture is an RNN (and hence conceptually, a new hidden state vector h_{t} is introduced) to handle adaptation for different given tasks, whereas (Janner, 2019) used a probabilistic feed-forward network as the model.

Experiments are conducted on standard RL meta-learning benchmarks (e.g. ForwardBackward Halfcheetah), to compare sample efficiencies between different model-based techniques.

### **Strengths**
- Contributes to the literature in model-based techniques in the meta-learning setting, which is generally lacking.
- Experimentation seems to show the desired result - i.e. the paper's method is more sample efficient than both PEARL and L2A.
- Paper shows that a small modification of (Janner, 2019) using an RNN can solve meta-learning problems in RL.

### **Weaknesses**
- A bit _too_ similar to (Janner, 2019). Novelty-wise, it seems like a small (but effective) modification of the work in (Janner, 2019) from model-based optimization in the single MDP setting to the meta-learning setting, by including a recurrent/RNN model. The main contribution seems to be this modification, but all other techniques of the paper (e.g. defining the correct terms such as a _branched rollout_, generating the concepts such as C(eps, eps'), etc.) are exactly the same in (Janner 2019). I think the paper overemphasizes the portions that have already been written in (Janner, 2019).

- A bit hard to read at times, due to the denseness of the notation. This is especially encountered in page 6, where there are large chunks of text + math without breathing room. I suggest that the authors provide a high level summary in the main body, especially because most terms have already been defined in (Janner, 2019).

### **Clarity Comments**
- Since the paper is very similar to (Janner, 2019), it may be useful to highlight the main differences (e.g. different coloring of changed lines in the Algorithmic boxes), rather than redefining everything already mentioned - it was difficult to parse which sections were novel/changed and which were from (Janner, 2019).

Overall, the main benefit seems to be this RNN modification and relevant meta-learning experiments (which is why I give a marginal accept), but I do think that the authors can re-write the paper in a much simpler fashion.

---

> ### Author Response · Authors · 2020-11-19
> **Our response to Reviewer 5's review**
>
> Thank you very much for your comments.
> We posted our response to common concerns on our work, at the top of this board.
>
> Here is our response to your comments:
>
> > A bit too similar to (Janner, 2019). Novelty-wise, it seems like a small (but effective) modification of the work in (Janner, 2019) from model-based optimization in the single MDP setting to the meta-learning setting, by including a recurrent/RNN model. The main contribution seems to be this modification, but all other techniques of the paper (e.g. defining the correct terms such as a branched rollout, generating the concepts such as C(eps, eps'), etc.) are exactly the same in (Janner 2019). I think the paper overemphasizes the portions that have already been written in (Janner, 2019).
>
> We would like to highlight the following theoretical analysis as other main contribution:
> We improve the theorems proposed in Janner et al. (2019) to bound the performance of branched rollouts more properly.
> Specifically, in Theorems 4.2 and 4.3 proposed in Janner et al. (2019), some important premises (e.g., multiple model-based rollouts factors) are not taken into account.
> The oversights of those important premises induce a large mismatch between those for their theorems and those made for the actual implementation of branched rollouts.
> The mismatch becomes larger especially when the model-rollout lengths (k) is large (more precisely, the mismatch is incurred in all cases where $k>1$); namely, Theorems 4.2 and 4.3 do not properly bound the performance of the branched rollouts in large $k$.
> Properly bounding the performance of the branched rollouts in large $k$ is important for correctly comparing with other rollouts (e.g., full rollouts) or estimating the optimal $k$ for example.
>
> In this paper, we take the premises into account and propose refined theorems (Theorems 4 and 5 in A.7 in our paper). The refined theorems do not have the issue of Janner et al. (2019), and enable us to bound the performance in large k more properly.
> In addition, we extend our Theorems 4 and 5 rather than Theorems 4.2 and 4.3 in Janner et al. (2019) when we derive the theorems for the POMDP case (Theorems 2 and 3 in our paper).
>
>
> > A bit hard to read at times, due to the denseness of the notation. This is especially encountered in page 6, where there are large chunks of text + math without breathing room. I suggest that the authors provide a high level summary in the main body, especially because most terms have already been defined in (Janner, 2019).
>
> Thank you for the suggestion. We will improve the clarity of the paper as suggested.

---

### Author Response · Authors · 2020-11-19
**Our response to common concerns on our work**

Dear reviewers,
Thank you very much for your thoughtful review comments.

We understand that the reviewers are mainly concerned about novelty and clarity.
To improve clarity, we will revise the paper according to the review comments.
In this reply and the first revision, we focus on clarifying our novelty and contributions and answering the questions raised in the reviews.

We understand that our work can be seen as a POMDP extension of known theories and algorithm.
However, we would like to argue that this is not a critical weakness of the paper since there are a number of prominent and important works that have extended the theory and algorithms known in MDP to the ones for POMDP (e.g., [1] and [2]).
We hope that the reviewers will re-evaluate the strengths and weaknesses of our work by taking this fact into account.

Further, we would like to emphasize our contribution on the meta-RL frontier: "Our work is the first attempt to enable sample efficient meta-RL using the model-based framework with a theoretical guarantee."
As we introduced in Section 1, in the previous model-based meta-RL literature, it has not been clear how learning the meta-model relates to real environment performance.
We provide relations between them (Theorems 1, 2 and 3) and build the model-based meta-RL framework on the basis of the theories.
We also show that M3PO achieves better sample efficiency than existing meta-RL methods.

In addition, in Section 4, we provide a discussion about how existing meta-RL settings can be recovered in our setting. This discussion indicates that our framework and theories can be flexibly applied to a variety of existing meta-RL settings.
Therefore, we believe that our framework and theory is useful in the meta-RL community. (For example, one may extend our framework and theories by introducing additional assumptions on their target meta-RL setting.)


[1] David Silver, Joel Veness, Monte-Carlo Planning in Large POMDPs, NeurIPS, 2010.
[2] Matthew Hausknecht, Peter Stone, Deep Recurrent Q-Learning for Partially Observable MDPs. AAAI, 2015

---

### Decision · Program_Chairs · 2021-01-07
**Final Decision**

**Decision:**

Reject

**Comment:**

The paper presents a meta-learning for Model-based RL that introduces branched rollouts to improve sample efficiency of the learned model.

While the paper addresses an important topic of sample efficiency in RL, and provides theoretical analysis, the reviewers raised concerns with the novelty and clarity. The extension to POMDP setting is certainly important technological contribution, albeit a straightforward. To be suitable for publication the work needs to make stronger case for the significance of the method.